# OPTIMAL FORMATS FOR WEIGHT QUANTISATION

## ABSTRACT

Weight quantisation is an essential technique for enabling efficient training and deployment of modern deep learning models. However, the recipe book of quantisation formats is large and formats are often chosen empirically. In this paper, we propose a framework for systematic design and analysis of quantisation formats. By connecting the question of format design with the classical quantisation theory, we show that the strong practical performance of popular formats comes from their ability to represent values using *variable-length codes*. We frame the problem as minimising the KL divergence between original and quantised model outputs under a model size constraint, which can be approximated by minimising the squared quantisation error, a well-studied problem where entropy-constrained quantisers with variable-length codes are optimal. We develop non-linear quantisation curves for block-scaled data across multiple distribution families and observe that these formats, along with sparse outlier formats, consistently outperform fixed-length formats, indicating that they also exploit variable-length encoding. Finally, by using the relationship between the Fisher information and KL divergence, we derive the optimal allocation of bit-widths to individual parameter tensors across the model's layers, saving up to $0.25$ bits per parameter when applied to large language models.

## 1 INTRODUCTION

Weight quantisation enables large deep learning models to run on low-resource hardware and edge devices by saving space and memory bandwidth usage. It can be seen as an optimisation problem, where the goal is to retain the behaviour of the high-precision reference model while reducing the total number of bits needed to store its parameters. This naturally splits into two sub-problems of format design and quantisation procedure, both of which are highly active areas of research. We focus on the format design question, i.e., how to choose a representation space for model parameters. This is somewhat independent from the quantisation procedure, which aims to find an optimal point in that space. The space of possible formats for a model is rich, where element formats including integer, floating-point and non-uniform quantisers can be combined with tensor, channel or block scaling and augmented using sparse outlier storage or rotations (Dettmers et al., 2022a; Dettmers & Zettlemoyer, 2023; Dettmers et al., 2023; Dotzel et al., 2024; Rouhani et al., 2023; Tseng et al., 2024a). Since this combinatorial space is too large to be explored directly, empirical studies typically narrow the search space.

This work addresses the problem of optimal format design, minimising the KL divergence against a reference model under a memory constraint. Following Kim et al. (2024), who observe that Fisher information can be used to predict the degradation based on the squared error of quantisation, we employ classical quantisation theory (Panter & Dite, 1951; Lloyd, 1982) to derive optimal element-wise quantisers that minimise squared error. Given this scheme for element formats, we empirically evaluate scaled formats based on the absmax or RMS of the tensor, channel or sub-tensor block, for direct-cast quantisation and quantisation-aware training. Our main takeaway is that *the most effective quantisation formats all employ some form of variable-length encoding* (see Figure 1). Block formats outperform optimal elementwise formats by effectively allocating their scale bits to represent the block maximum. Tensor-scaled formats can be effective if sparse outliers are stored separately, again implying variable bits-per-element. Finally, quantisation followed by lossless compression employs variable length explicitly and doesn't benefit from block scaling or sparse outlier removal.

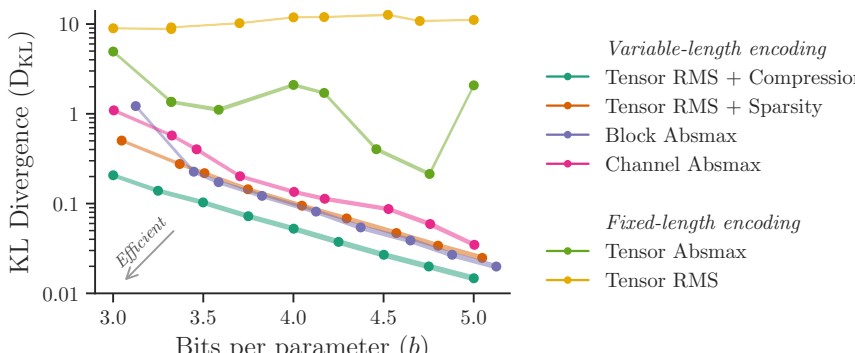

Figure 1: The trade-off between average bits per parameter and top-$k$ KL divergence for Llama 3.1 8B. To approach optimal performance, some form of variable-length encoding is needed: lossless compression, block (or channel) absmax scaling or $0.1\%$ sparse outlier removal. The shaded line width is $\pm 2$ standard error over evaluation data. See Figure 8 for other models.

**Contributions** Our work explains the performance of various quantisation schemes through the lens of optimising KL divergence and weight statistics. Towards this, we propose $\sqrt[3]{p}$ block-scaled Normal, Laplace and Student-t non-uniform quantisers. We also propose the signmax scaling scheme. These are, to the best of our knowledge, novel contributions.

**Outline** Section 2 presents our framework: the optimisation problem, assumptions and solutions. Experiments on synthetic data in Section 3 and LLM quantisation in Section 4 demonstrate that block and sparse outlier formats outperform fixed-length codes. Sections 5 to 7 present related work, limitations and conclusions. Table 3 provides a reference for our notation.

## 2 OPTIMAL QUANTISATION FORMATS

In this section, we present our framework for format design. We begin by defining the overall optimisation problem at the model level, showing how this objective can be reduced to minimising the squared error of individual tensors through appropriate approximations. Next, we present solutions for RMS and absmax scaling schemes, as well as a lossless compression scheme. Finally, we revisit model-level optimisation by proposing a scheme for optimal bit allocation across tensors.

Consider a probabilistic model with output $y$ conditioned on input drawn from a dataset $x \sim \mathcal{X}$, where the pdf $\mathrm{p}_\theta(y \mid x)$ is represented by a neural network, parametrised by $\theta \in \mathbb{R}^{|\theta|}$. We wish to find a quantised parameter vector $\tilde{\theta}^\star$ which is in $\tilde{\Theta} \subset \mathbb{R}^{|\theta|}$, minimising the expected KL divergence of $\mathrm{p}_{\tilde{\theta}}(y \mid x)$ against the reference model,

$$\tilde{\theta}^\star := \underset{\tilde{\theta} \in \tilde{\Theta}}{\arg\min}\, \mathrm{D}_{\mathrm{KL}}\left(\mathrm{p}_\theta \| \mathrm{p}_{\tilde{\theta}}\right),$$

$$\mathrm{D}_{\mathrm{KL}}\left(\mathrm{p}_\theta \| \mathrm{p}_{\tilde{\theta}}\right) := \underset{x \sim \mathcal{X}}{\mathbb{E}}\left[\underset{\mathrm{p}_\theta(y|x)}{\mathbb{E}}\left[\log \frac{\mathrm{p}_\theta(y \mid x)}{\mathrm{p}_{\tilde{\theta}}(y \mid x)}\right]\right], \tag{1}$$

$$\text{with } |\tilde{\Theta}| \leq 2^{|\theta| \cdot b}.$$

The final line induces a compression constraint on $\tilde{\Theta}$, specifying an average of $b$ bits per parameter. Considering $\tilde{\theta}$ close to $\theta$, we can approximate the objective as

$$\mathrm{D}_{\mathrm{KL}}\left(\mathrm{p}_\theta \| \mathrm{p}_{\tilde{\theta}}\right) \approx \frac{1}{2}(\tilde{\theta} - \theta)^\top F (\tilde{\theta} - \theta), \tag{2}$$

where $F \in \mathbb{R}^{|\theta| \times |\theta|}$ is the Fisher information matrix of the reference model (see Section A for a derivation). As a further simplification, we assume the cross terms are small, in which case the approximation reduces to the squared quantisation error, weighted by the diagonal of the Fisher information matrix.

Deep learning model parameters can be partitioned into *tensors*, often the maximal sets of parameters that can be applied in parallel to the intermediate activations — in other words, where the input to the forward pass operation of any scalar parameter does not depend on any other parameter from the same tensor. It is convenient to encode each tensor using a single format, although different tensors may use different formats, so we expand $\tilde{\Theta} = \prod_t \tilde{\Theta}_t$, and $\tilde{\theta}_{T_t} \in \tilde{\Theta}_t$ where $T_t \in \mathbb{N}^{|T_t|}$ is a vector of parameter indices belonging to tensor $t$. For much of our analysis, we further approximate the diagonal of the Fisher matrix as constant ($= \bar{f}_t$) within each parameter tensor. With this approximation, the increase in KL divergence due to quantisation is a weighted sum across parameter tensors of the unweighted squared error,

$$D_{\mathrm{KL}}\left(p_\theta \| p_{\tilde{\theta}}\right) \approx \frac{1}{2} \sum_t \bar{f}_t \cdot \sum_{i \in T_t} (\tilde{\theta}_i - \theta_i)^2. \tag{3}$$

This form is convenient, as the squared error is easy to compute and the diagonal of $F$ can be estimated with computational and memory costs comparable to a few steps of SGD. We test this equation for predicting the KL divergence after perturbing with iid normal noise in Figures 11 and 13.

## 2.1 Optimal tensor formats for known distributions

We now turn to the problem of designing a format to represent a single parameter tensor. The formats we consider all operate on blocks of data — all or part of the tensor. The *block size $B$* is fixed within a tensor, but may vary across tensors. For the $i$th block of the $t$th tensor, we wish to quantise the parameters sub-vector $\theta_{C_{ti}}$, given block indices $C_{ti} \in \mathbb{N}^B$, but to aid readability we will drop the indices and use $\theta$ directly. Using the approximate relationship given by Equation (3), KL divergence is minimised by minimising the squared reconstruction error of each parameter. We therefore consider the following optimisation problem for a block of parameters $\theta \in \mathbb{R}^B$:

$$\text{find: } \mathtt{quantise}: \mathbb{R}^B \to Q \ , \ \mathtt{dequantise}: Q \to \mathbb{R}^B \tag{4}$$

$$\text{to minimise: } E^2 := \sum_{i \in [1..B]} \left(\theta_i - \mathtt{dequantise}(\mathtt{quantise}(\theta))_i\right)^2$$

$$\text{such that: } |Q| \leq 2^{B \cdot b},$$

i.e. the set of quantised representations $Q$ is subject to a compression constraint of $b$ bits per element in the block of $B$ elements.

**Cube root density quantiser** For scalar quantisation ($B = 1$) of samples from a known pdf $p^{\mathcal{D}}$, this optimisation problem has a classical solution (Panter & Dite, 1951). The set $Q^{\mathrm{elem}} \subset \mathbb{R}$ of non-linear codepoints is distributed with density proportional to the cube root of the pdf, and quantisation rounds to the nearest codepoint:

$$\mathtt{quantise}^{\mathrm{elem}}(\theta) = \mathrm{argmin}_{q \in Q^{\mathrm{elem}}} |\theta - q|, \quad \mathtt{dequantise}^{\mathrm{elem}}(q) = q.$$

When $\mathcal{D}$ is Normal, Laplace or Student-t, we observe that $\mathtt{density}(Q^{\mathrm{elem}}) \propto \sqrt[3]{p^{\mathcal{D}}}$ is proportional to the pdf of $\mathcal{D}'$, the same parametric distribution as $\mathcal{D}$ but with different parameters. Therefore $Q^{\mathrm{elem}}$ can be derived from the inverse cdf of $\mathcal{D}'$, see Section B.2 for details and E for code examples.

**Linear scaling** Linear scaling schemes combine a scalar element format with a shared (per-block) scale format. The block statistic $\mathtt{norm}(\theta)$ is used to scale the data before quantisation and is stored alongside the data for the sake of dequantisation:

$$Q^{\mathrm{linear}} := \mathbb{R} \times (Q^{\mathrm{elem}})^B \ , \ \mathtt{norm}: \mathbb{R}^B \to \mathbb{R}$$

$$\mathtt{quantise}^{\mathrm{linear}}(\theta) = \left[n, \ \mathtt{quantise}^{\mathrm{elem}}\left(\frac{\theta_i}{n}\right)_{\forall i \in [1..B]}\right] \quad \text{where } n = \mathtt{norm}(\theta)$$

$$\mathtt{dequantise}^{\mathrm{linear}}(n, q)_i = n \cdot \mathtt{dequantise}^{\mathrm{elem}}(q_i).$$

Note that these equations cannot yet obey the compression constraint, as $Q^{\mathrm{linear}}$ is uncountable; for a practical scheme we must instead store $\mathtt{quantise}^{\mathrm{scale}}(n)$ using an appropriate format. We now consider three choices for $\mathtt{norm}$, corresponding to RMS, Absmax and Signmax scaling formats.

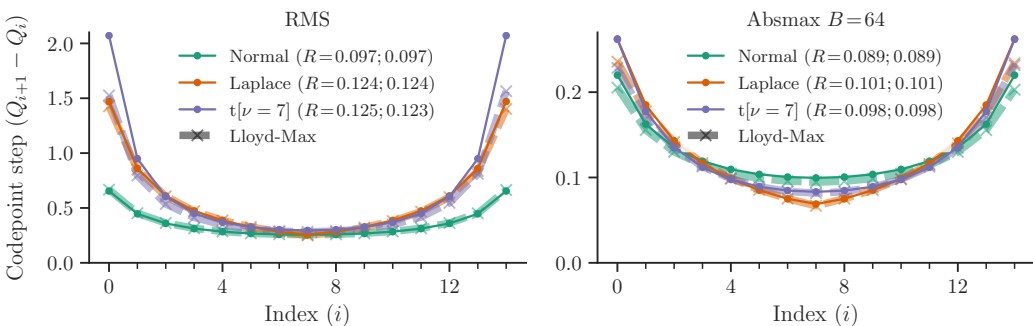

Figure 2: 4-bit quantisation curve gradients for Normal, Laplace, Student-t distributions, showing strong agreement between cube root density and Lloyd-Max (k-means), which optimises codepoints against samples directly, without assumptions on the underlying distribution. The legend shows relative quantisation error $R$ for "(cube root quantiser; Lloyd-Max)" for data matching the quantiser. *(Left)* RMS-scaled formats. The cube root density rule breaks down with the heavy tails of Student-t. *(Right)* absmax-scaled formats. The discrepancy at the extremes occurs because the cube root quantiser has a special case for $\pm 1$, whereas Lloyd-Max treats it as a single distribution to quantise.

**RMS scaling**  Applying linear scaling with $\mathrm{norm}(\theta) = \sqrt{\frac{1}{B}\sum_i \theta_i^2}$, we obtain RMS scaling. If we replace the random variable $n$ with a point estimate at its expected value, then $\frac{\theta_i}{n}$ follows $\mathcal{D}^{\mathrm{elem}}$, a scaled version of $\mathcal{D}$ such that $\mathbb{E}\left[\theta_i^2\right] = 1$. Moment matching of the RMS can provide the parameters of $\mathcal{D}^{\mathrm{elem}}$ needed for an optimal format according to the cube root density rule (Table 4). See Figure 2 for example quantisation curves.

**Absmax scaling**  Linear scaling with $\mathrm{norm}(\theta) = \max_i |\theta_i|$ gives absolute-maximum scaling, a popular block quantisation scheme. In this case $\mathcal{D}^{\mathrm{elem}}$ has support $-1 \le \theta_i \le 1$. Following Yoshida (2023), we consider $\mathcal{D}^{\mathrm{elem}}$ as a mixture of two components: (1) the normalised maximum value, which is a transformed Bernoulli distribution and (2) the normalised distribution of everything else in the block, which was not the maximum. To approximate (2), we observe empirically that the marginal distribution of $\theta_{i \ne \arg\max_j |\theta_j|}$ is a good match to a truncated $\mathcal{D}$, where the truncation point is the block maximum (Figure 15). We then use a closed form approximation to $\mathbb{E}\left[\max_i |\theta_i|\right]$ (Table 4) to calculate the truncation points. To construct $Q^{\mathrm{elem}}$, we always include $\pm 1$, then use the inverse cdf of the truncated-and-scaled $\mathcal{D}$ to distribute the rest of $Q^{\mathrm{elem}}$ according to the cube root rule. Example quantisation curves are shown in Figure 2.

**Signmax scaling**  Observing that the distribution of block-scaled data is well-approximated by a mixture of the maximum and non-maxima, it seems natural to also try *signmax* scaling. In this scheme, the block scale is set to the signed absolute maximum, $\mathrm{norm}(\theta) = \theta_{\hat{i}}$ where $\hat{i} = \arg\max_i |\theta_i|$. The element format can then assume that the maximum is always at $+1$ (not $\pm 1$) and allocate a pair of special codepoints $\{0, 1\}$ with the rest specified according to $\sqrt[3]{\mathrm{p}}$ (see Figure 3). This comes at the cost of requiring a sign bit for the block scale, i.e. $\frac{1}{B}$ bits per element.

**Symmetric/Asymmetric variants**  One important detail is the representation of zero. Practical implementations prefer an even number of codepoints, so allocating a codepoint for zero mandates asymmetry or waste. However, exact zero has been shown empirically to be valuable (Liu et al., 2025). The $\sqrt[3]{\mathrm{p}}$ scheme is easily adapted to provide symmetric and asymmetric variants for both RMS and absmax scaling (Figure 3). For block scaling the asymmetry is purely in resolution, while for RMS it provides both additional resolution and range on one side.

## 2.2 Unknown distributions

Since the underlying distribution of parameters of neural networks after training is unknown, we cannot apply the techniques described in Section 2.1 directly. One way forward is to assume a distribution family and select *scale* and *shape* (e.g. Student-t $\nu$) *distribution parameters* to fit the data. The scale can be fit using moment-matching (absmax or RMS) between the quantiser and parameters, or by explicit search to minimise $E^2$ (see Figure 23 for an example). We find explicit search for scale and shape to be more reliable than moment-matching. An alternative approach is

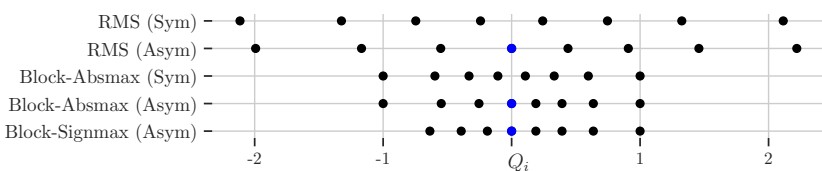

Figure 3: 3-bit $\sqrt[3]{\mathrm{p}}$ codepoint distributions for normally distributed data, illustrating RMS, absmax and signmax scaling methods and symmetric/asymmetric variants (with $B = 64$ for block formats). The principal benefit of asymmetric variants is that they have an encoding for **0**. INT formats are asymmetric, while most floating-point formats are symmetric but represent zero twice ($\pm 0$).

to directly optimise a non-uniform format against the parameters to solve Equation (4). A standard solution is the Lloyd-Max algorithm (Lloyd, 1982; Max, 1960), i.e. 1D $k$-means. Both distribution parameter search and Lloyd-Max can solve a weighted objective, so we can drop the scaled-identity approximation to the Fisher information, using $\mathrm{diag}(F)$ as a weight on the importance of each parameter, as proposed by Kim et al. (2024).

### 2.3 OPTIMAL TENSOR FORMATS WITH COMPRESSION

An alternative approach is to first use a (lossy) quantiser such as those described in Section 2.1 then follow it with a lossless compressor, operating on quantised data in $Q$. In this case we can optimise the quantiser according to Equation (4), replacing the last line with an entropy constraint:

$$\ldots \text{such that: } \mathbb{E}\left[\mathcal{I}(\texttt{quantise}(\theta))\right] \leq B \cdot b$$

$$\text{where: } \mathcal{I}(q) = -\sum_{i \in [1..B]} \log_2 \mathrm{p}^{\mathcal{Q}}(q_i).$$

I.e. $\mathcal{I}(q)$ computes the information content of a quantised block under a model of their values given by $\mathrm{p}^{\mathcal{Q}}$ and we assume an optimal compressor approaching the Shannon (1948) limit. Practical Huffman codes (Huffman, 1952) or arithmetic coding (Witten et al., 1987) can approach this limit.

**Compressed grid** Under this new constraint if $B = 1$ the optimal distribution of elementwise codepoints, assuming "high rate", is a uniform grid i.e. $\texttt{density}(Q^{\mathrm{elem}}) = \text{const}$ (Gish & Pierce (1968), see also Section B.3). The probability model $\mathrm{p}^{\mathcal{Q}}$ for compression can either be estimated based on samples or derived by transforming $\mathcal{D}$ by $\texttt{quantise}(\theta)$, which in the case of elementwise quantisers is trivial: via the cdf or approximately via the pdf of $\mathcal{D}$.

### 2.4 OPTIMAL BIT-WIDTH ALLOCATION

We have seen that Fisher information can predict KL divergence due to quantisation and further observe that the average Fisher information varies substantially across tensors (see Figure 12). This suggests that there may be an optimal *variable* allocation of bits across parameter tensors, while respecting the average bit-width constraint at the model level. This scheme should allocate more bits to tensors that are more "sensitive", i.e., having higher Fisher information. Using the asymptotic optimal quantisation error of Zador (1982), we derive the following *variable bit allocation* scheme:

$$b_t^\star := b^0 + \log_2 \mathrm{RMS}(\theta_{T_t}) + \frac{1}{2} \log_2 \bar{f}_t, \tag{5}$$

where $b_t^\star$ is the bit width of tensor $t$, $\bar{f}_t$ is the average of the Fisher matrix diagonal and $b^0$ is chosen to satisfy the overall size constraint. Intuitively, if tensor a has $4\times$ the Fisher information of tensor b then a uses 1 more bit than b. See Figure 17 for an example and Section B.5 for the derivation.

## 3 ANALYSIS — SIMULATED DATA

Motivated by the parameter statistics of Figure 25, we consider $\mathcal{D} \in \{\text{Normal}, \text{Laplace}, \text{Student-t}\}$ in turn. Using the methods described above, we compare optimal formats with block absmax or tensor RMS scaling on iid data from each distribution. Our aim is to establish whether there are benefits to block absmax formats for iid data, and where those benefits come from. Unless noted, all

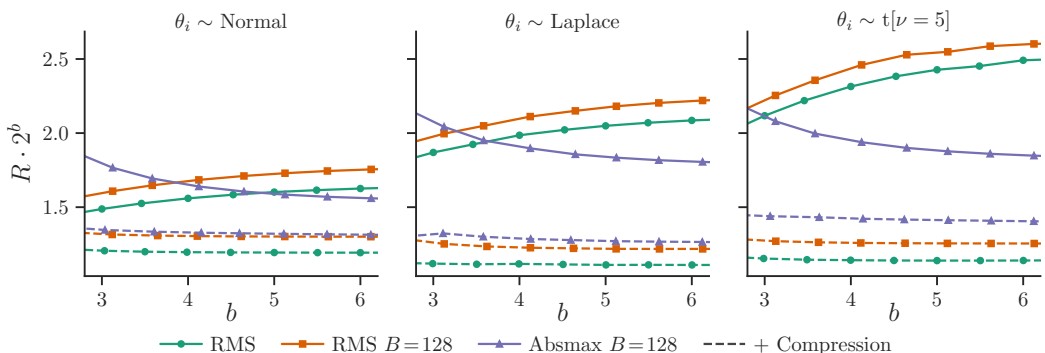

Figure 4: The error/size tradeoff for different data distributions (column) and optimal quantisers (hue). Surprisingly, block absmax quantisers can outperform tensor RMS formats for iid data, even though there is no inherent block structure. However this situation is reversed when adding optimal compression, implying that block absmax quantisers exploit some form of variable-length coding.

experiments in this section use `bfloat16` block scale and symmetric codepoint distributions. We report $R$, the ratio of RMS error to parameter RMS and often plot $R \cdot 2^b$ to make diagonal error/bits trade-off lines horizontal. See Section C for further details.

*Preliminaries* To validate the cube root rule, we test a generalised quantiser with pdf exponent $\alpha$ in Figure 22, finding that the cube root setting ($\alpha = \frac{1}{3}$) performs best and similarly to Lloyd-Max, outperforming quantile quantisation ($\alpha = 1$). For block scaled formats, we must choose an appropriate block size. Smaller blocks have lower error from a tighter block range but incur greater space overhead from storing the block scale. Figure 21 shows that $B = 128$ is a good choice for these distributions. The figure also validates our default choice of `bfloat16` over E8M0.

**Block formats exploit variable-length encoding.** Our main result is the tradeoff between error and bit width for various scaling strategies with optimal quantisers, Figure 4. We were surprised to find that block absmax formats can outperform tensor RMS formats that use optimal element quantisers, even for iid simulated data. However when adding compression, using a variable number of bits to encode each value, the advantage of block formats disappears and tensor RMS scaling performs best. This suggests a perspective on the benefit of block absmax formats — instead of viewing them as a way to avoid clipping, we can view them as a variable-length code, using additional (scale) bits to encode the block maximum. Since they outperform optimal fixed-length codes, they must somehow exploit variable-length coding. While the exact mechanism isn't clear, we provide an illustrative depiction of the effective code length in Figure 5.

*Additional observations* We compare standard formats against optimal block element formats in Figure 18. Here we see that NF4 (Dettmers et al., 2023) is not optimal for RMS error across different block sizes, and that E2M1 is consistently better than `INT4` and `E3M0`. For floating-point formats, Figure 19 shows that the optimal number of exponent bits generally doesn't change as the total bit allocation grows. This is expected, since exponent bits govern the shape of the quantisation density function while mantissa bits govern the resolution, and the optimal shape should remain fixed. Figure 20 shows the benefit of using 4-10 scale mantissa bits over E8M0. In Figure 24, we see that an elementwise Huffman code approaches the theoretical compression performance.

## 4 EXPERIMENTS

We evaluate a wide variety of weight formats described above for quantisation of pretrained language models from the Llama 3, Qwen 2.5, Gemma 3 and Phi 4 families. To enable a broad evaluation, most results use direct-cast quantisation, sometimes called round-to-nearest, which is a simple quantisation technique that performs one-shot conversion, without using data or fine-tuning. The primary comparison is an efficiency trade-off between top-$k$ KL-divergence of quantised and original model predictions, $D_{KL}$, against average bits per parameter, $b$. We also use $\rho := D_{KL} \cdot 2^{2b}$ as a measure of inefficiency of representation. See Section D for further details of our methodology.

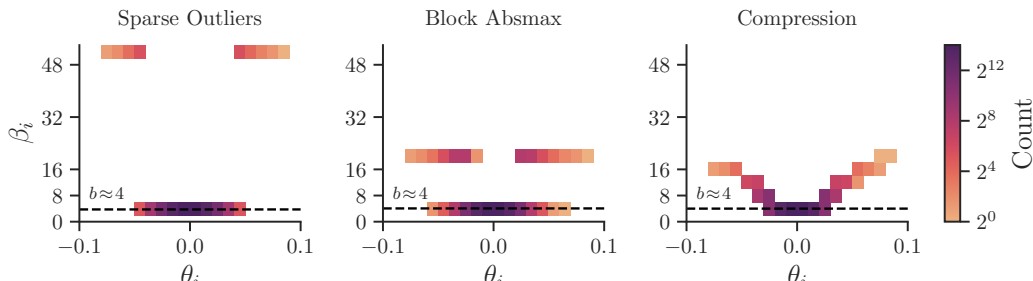

Figure 5: A 2D histogram of bits $\beta_i$ used to encode parameter $i$ from the first MLP down-projection from Llama 3.1 8B, illustrating how different schemes achieve variable-length encoding. *(Left)* sparse outliers create a distinct step between regular values and the $0.1\%$ largest absolute values. *(Center)* block absmax can be seen as using the `bfloat16` scale to represent the block maximum and fewer bits for everything else. The histogram has an overlap, since the maximum is per-block not global. *(Right)* lossless compression on a uniform grid with $\beta_i = -\log_2 p_i$, where $p_i$ is the proportion of parameters assigned to that quantisation bucket.

**Uniform quantisation with compression is efficient.** Figures 1 and 8 show that a uniform grid quantiser followed by optimal lossless compression consistently outperforms other approaches. With compression, tensor RMS scaling is sufficient and can be folded into the grid resolution. Huffman coding achieves near-optimal compression, as shown in Figure 8.

**Variable-length encoding is necessary for good performance.** Figures 1 and 8 also demonstrate the characteristics of near-optimal formats without lossless compression. All employ block or channel absmax scaling and/or separate storage of sparse outliers (Kim et al., 2024). Our search over a wide range of element formats was unable to find fixed-length schemes that can reach the same performance as block or sparse formats. Consistent with our observations of Section 3, this indicates that they exploit variable-length encoding. We also observe that there is no benefit in adding block absmax scaling or sparse outlier removal to lossless compression (see Figure 28), indicating that their benefit comes from the same source. Alternatively, random rotations can mitigate the poor performance of fixed-length codes Figure 29, but cannot match the performance of optimal variable-length codes.

**Variable bit allocation improves efficiency.** The variable bit allocation scheme promises to reach the same overall compression level for a model with less degradation by allocating more bits to parameters with higher Fisher information. Figure 6 shows that this is indeed the case, with a strong improvement across 8 of 11 models and different formats. The exception is Gemma models, where our prediction of KL based on Fisher information also breaks down (Figure 13).

**Downstream tasks & QAT are consistent with KL divergence & direct-cast.** As direct-cast quantisation is known to be suboptimal, we confirm that the ranking of headline formats is broadly consistent after quantisation-aware training (QAT) and when evaluating on a suite of downstream tasks in Figure 7, with a side-by-side comparison against direct-cast results in Figure 9. Downstream performance saturates, so that QAT has the largest advantage at $b \leq 4$ and format selection is most important at $b \approx 3$ bits. See Tables 1 and 2 for results on individual tasks.

*Additional observations* In Figure 31, we compare element formats against a Student-t baseline over $b \in [3..5]$, indicating the Student-t format performs best in almost all cases. Figure 32 compares 4-bit $\sqrt[3]{p}$ formats against NF4 and SF4 baselines: $\sqrt[3]{p}$ Normal and NF4 show similar performance, but $\sqrt[3]{p}$ Student-t outperforms SF4. For block absmax scaling, Figure 33 confirms the results from simulated data — a block size near 128 and 4-7 scale mantissa bits perform best. We compare symmetric, asymmetric and signmax variants for block scaling with integer or Student-t element formats in Figure 34, finding that signmax delivers a consistent improvement across models and particularly for small $b \approx 3$. The performance of symmetric vs asymmetric variants is inconsistent across models. In Figure 35 we evaluate different ways to choose the quantiser scale. Search to minimise $R$ is better than moment matching when using RMS scaling, but can be harmful for absmax scaling unless weighted by the per-parameter Fisher information.

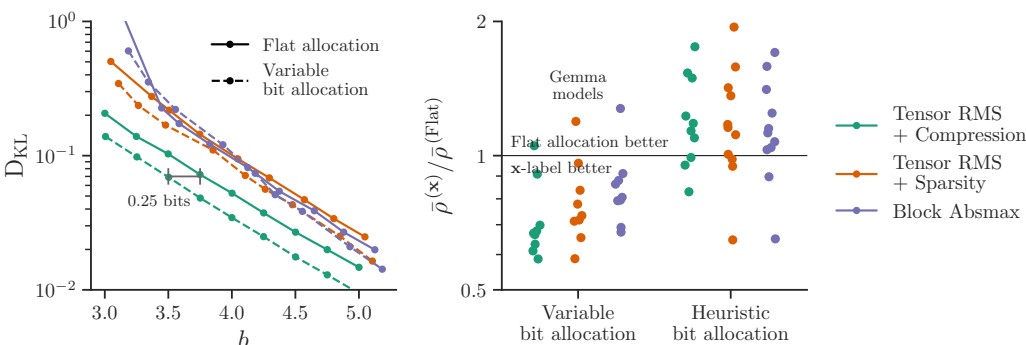

Figure 6: The performance of the Fisher-based variable bit allocation scheme of Equation (5). *(Left)* the tradeoff curve for Llama 3.1 8B, showing a general shift to the left, although some settings for absmax scaling are degraded. *(Right)* the average scaled KL of different bit allocation schemes compared with flat allocation, improving all models except for Gemma 3 4B and 12B, and Qwen 2.5 3B using Block Absmax (some of which lie outside the plotted range). See Figure 30 for a cross-domain result and an explanation of *heuristic bit allocation*.

## 5 RELATED WORK

We separate the research of weight quantisation into two broad categories: the design of *numerical formats* with which to represent the model parameters, and *quantisation techniques* for adapting the parameters in order to minimise the quantisation error.

**Numerical formats** Low-precision formats have been widely utilised for training and inference of models, with the half-precision `bfloat16` format (Wang & Kanwar, 2019) becoming the de facto standard. Modern accelerator hardware offers further support for lower-precision floating point formats, such as `fp8` (Wang et al., 2018; Noune et al., 2022), `fp4` (Sun et al., 2020) and `fp6` (Gernigon et al., 2023). Dettmers & Zettlemoyer (2023) compare the effectiveness of different formats for inference, showing that 4-bit precision can be optimal on the accuracy/size trade-off. Similarly, Liu et al. (2025) show that optimising the training scheme further highlights the potential of even lower bit widths, such as 2-bit quantisation. Low-precision formats are often accompanied with block scaling techniques to improve the accuracy, such as in the proposed MX formats (Rouhani et al., 2023), where blocks of integer or floating point elements are combined with a per-block scale element. As transformer inference is often bottlenecked by memory transfers, formats without hardware arithmetic support can be useful for compression, for example using a look-up table (LUT) for dequantisation. Dettmers et al. (2023) introduce `NF4`, aimed to be the theoretically optimal 4-bit format under the assumption of normally-distributed parameters; similarly, Dotzel et al. (2024) introduce `SF4`, assuming a Student-t distribution. In both approaches, the authors derive the codebook so that each quantisation bin is equally populated under the assumed distribution. However, this does not lead to optimal codes in terms of the RMS error, which we instead motivate and pursue in our current work. Yoshida (2023) identifies that block size can have a significant impact on the scaled distribution, and derives `AF4` assuming a normal distribution. `AF4` is similar to our proposed block absmax $\sqrt[3]{p}$ Normal format, but optimises for absolute rather than squared error and uses a different approximation for the block maximum. Alternatively, a format's codebook can be fitted to a parameter tensor or a sub-tensor, typically by applying the Lloyd-Max algorithm (Lloyd, 1982; Max, 1960) to find the optimal codepoints (Zhang et al., 2018; Kim et al., 2024).

*Outliers, Rotations & Compression* Many techniques choose to handle outlier values separately, by storing them in higher precision. In `LLM.int8()` (Dettmers et al., 2022a), the authors identify outlier feature dimensions which they keep in higher precision, while storing the rest of the values in `int8`. Similarly, in SpQR (Dettmers et al., 2024) and SqueezeLLM, outlier parameter values are separately stored in higher precision. Alternatively, random or trained rotations can be used to suppress outliers and aid quantisation (Tseng et al., 2024a; Ashkboos et al., 2024; Liu et al., 2024b). Lossless compression has also been incorporated in Deep Compression and DFloat11 (Han et al., 2016; Zhang et al., 2025), which use Huffman codes to improve compression after quantisation.

*Vector Quantisation (VQ)* Instead of quantising each parameter independently, VQ (Linde et al., 1980) considers a block of parameters together. This can bring the benefits of a variable-length scalar

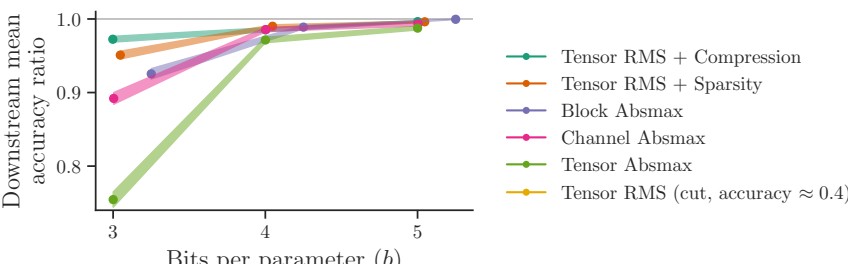

Figure 7: The trade-off between average bits per parameter and average downstream task performance for Llama 3.1 8B after QAT. Shaded lines show $\pm 2$ standard error over the evaluation data. Compared against Figure 1, the ranking of formats is broadly preserved at $b = 3$, but performance quickly saturates with $b \geq 4$. See also Figure 9 and Table 2.

code with a fixed-length vector code. VQ can also reduce bit-alignment overhead versus scalar codes and exploit correlations, but require careful design to constrain the size of the codebook. VQ has been successfully applied to parameter quantisation (Egiazarian et al., 2024; Tseng et al., 2024a;b).

*Sensitivity-aware formats* SqueezeLLM (Kim et al., 2024) uses Fisher information "FI" with weighted Lloyd-Max to select quantisation centroids. CherryQ (Cui & Wang, 2024) use FI to select a sparse set of parameters to retain in high-precision. Radio (Young, 2025) uses FI to select a bit-width per tensor or block of parameters, differing from our proposal by using an iterative scheme that does not assume locally constant FI. Choi et al. (2017) combine many of the ideas discussed here, using Hessian-weighted entropy-constrained scalar quantisation (ECSQ) to optimise codebooks before compressing with Huffman codes. Alternatively, techniques such as EvoPress (Sieberling et al., 2025) perform an explicit search over bit widths across different tensors.

**Quantisation techniques** These can be divided into two categories: quantisation-aware training (QAT) and post-training quantisation (PTQ). In QAT, the model is trained using backpropagation to optimise the quantised parameters for minimal end-to-end degradation (Yin et al., 2019; Liu et al., 2024a). Alternatively, PTQ techniques optimise the parameters without additional end-to-end training (Frantar et al., 2022). Such techniques are generally applicable to any chosen format, so can be used in with the approach presented in our work to further improve the accuracy of the model.

## 6 LIMITATIONS

Our study focuses on quantisation under a compression constraint, without directly addressing compute efficiency. While non-linear formats such as the $\sqrt[3]{p}$ Student-t quantiser offer improved model fidelity, they provide fewer opportunities for hardware acceleration than standard integer or floating-point formats. In practice, fast decoding of novel formats depends on optimised implementations as well as hardware support, which are promising directions for future work. Our framework is based on minimising the error under direct-cast quantisation. Although we have verified that insights from direct-cast results carry through to QAT for headline formats, interactions with finer-grained design choices (e.g. symmetric vs. asymmetric formats) and PTQ methods may require further evaluation. Finally, our empirical evaluations are restricted to transformer LLMs of 0.5–14B parameters. However, the framework itself is more general, and can be applied to other architectures and sizes.

## 7 CONCLUSIONS

Framing weight quantisation as an optimisation problem highlights the importance of choosing the right compression constraint. Under a codebook length constraint, $\sqrt[3]{p}$ and Lloyd-Max quantisers are optimal. Under an entropy constraint, uniform quantisation followed by lossless compression is optimal. We have shown that both block absmax and sparse outlier formats can be viewed as forms of variable-length encoding, exploiting the advantage of the entropy constraint. For the format designer, this suggests opportunities to develop practical formats that further close the gap with lossless compression. For the format user, it suggests coherent constructions to use, for example, block absmax formats or tensor RMS formats with sparse outliers.

REPRODUCIBILITY STATEMENT

We describe our experimental setup in detail in Section D, with code provided at *(see supplementary materials)* and outlined in Section E. The code relies only on public model checkpoints and datasets available on HuggingFace (Wolf et al., 2020), ensuring reproducibility.

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

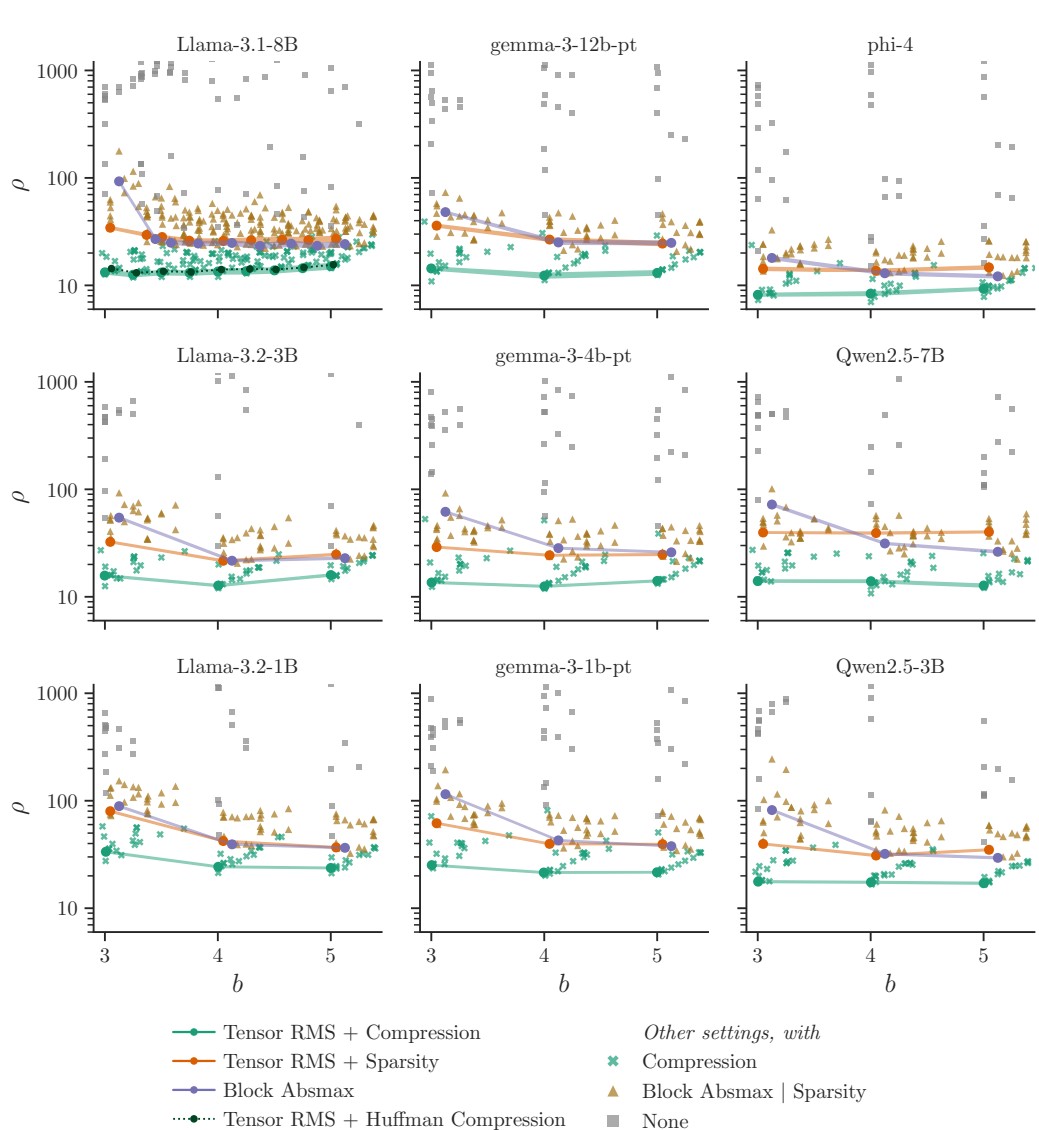

Figure 8: The trade-off between average bits per element and scaled top-$k$ KL divergence over different block scaling schemes (RMS, absmax), sparse outlier removal (on, off) and optimal lossless compression (on, off). This is similar to Figure 1, but on the vertical axis we show $\rho := \mathrm{D_{KL}} \cdot 2^{2b}$ to flatten the curve based on the error scaling limit of Zador (1982). We also show that simple per-element Huffman coding performs very close to the optimal compression which assumes the Shannon limit (Llama 3.1 8B only). Results are highly consistent across model families and sizes. Note: shaded lines show $\pm 2$ standard error over the evaluation data. Where trends appear noisy but error bars are tight (especially in Figure 1), this is due to the model itself — we are unable to quantify uncertainty due to model parameters, since there is only one independent fully-trained checkpoint for each family & size.

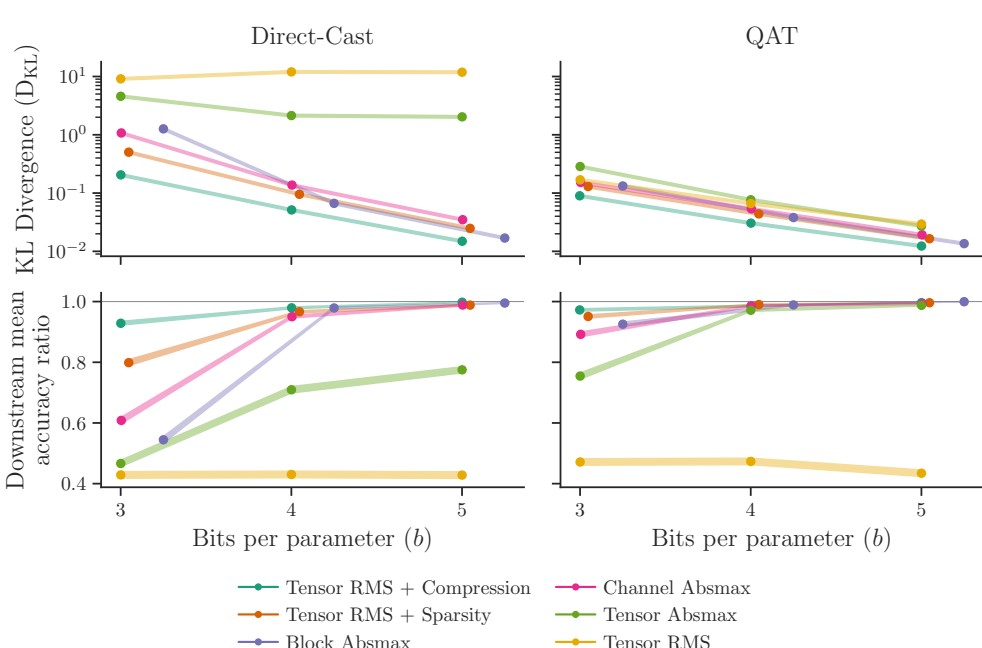

Figure 9: Comparison of formats using direct-cast *(Left)* and quantisation-aware-training *(Right)*, evaluated on validation KL divergence *(Top)* and downstream task average accuracy *(Bottom)*. QAT improves all results, as expected, but broadly preserves the ranking between the different formats. The ranking between formats on KL divergence and on downstream tasks is generally consistent, except for Tensor RMS scaling which is able to achieve good KL divergence after QAT, while remaining broken on downstream tasks. Downstream task performance saturates at large $b$, making format selection and QAT most impactful for $b = 3$.

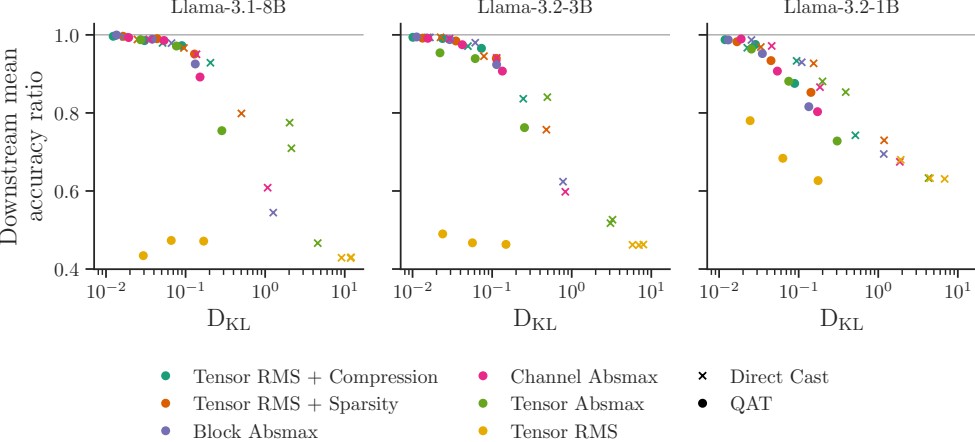

Figure 10: Correlation between validation KL divergence and average downstream performance for Llama models, for direct-cast quantisation and quantisation-aware-training, across a variety of formats and $b \in \{3, 4, 5\}$. There is generally a good correlation between KL divergence and downstream accuracy, with the notable exception of Tensor RMS scaling formats, which achieve low KL divergence after QAT without showing a corresponding improvement on downstream tasks.

Table 1: Downstream task results for headline formats with direct-cast quantisation at $b \approx 3$. Downstream tasks generally follow the KL divergence ranking.

| Format | $b$ | $D_{KL}$ | ARC-c | ARC-e | BoolQ | CSQA | HS | OBQA | PIQA | SIQA | WG |
|---|---|---|---|---|---|---|---|---|---|---|---|
| Baseline | 16.00 | 0.001 | 79.6 | 90.0 | 82.2 | 70.2 | 80.8 | 76.0 | 81.8 | 64.8 | 73.7 |
| Tensor RMS + C | 3.00 | 0.205 | 71.6 | 84.4 | 73.8 | 65.4 | 79.2 | 63.8 | 80.7 | 58.6 | 72.2 |
| Tensor RMS + Sp | 3.05 | 0.503 | 47.2 | 71.6 | 63.7 | 53.9 | 72.2 | 50.2 | 77.1 | 53.6 | 68.8 |
| Channel Absmax | 3.00 | 1.075 | 27.1 | 37.5 | 69.2 | 24.8 | 65.3 | 31.8 | 73.9 | 35.3 | 62.5 |
| Block Absmax | 3.25 | 1.264 | 24.4 | 33.0 | 62.0 | 21.9 | 46.1 | 27.0 | 70.4 | 37.1 | 59.2 |
| Tensor Absmax | 3.00 | 4.577 | 21.1 | 24.6 | 51.0 | 19.9 | 37.4 | 27.0 | 60.5 | 32.2 | 51.7 |
| Tensor RMS | 3.00 | 9.115 | 26.8 | 27.2 | 41.7 | 19.1 | 26.4 | 24.4 | 50.5 | 32.9 | 49.4 |

Table 2: Downstream task results using QAT at $b \approx 3$. There is a strong correlation between final validation KL and downstream task performance. Although QAT has improved all formats relative to direct-cast (Table 1), the ranking is consistent. The results are comparable to those of ParetoQ (Liu et al. (2025), Table 4). Note that baseline performance is different for our results under OLMES and that ParetoQ retains the embedding and final projection matrices in `bfloat16`.

| Format | $b$ | $D_{KL}$ | ARC-c | ARC-e | BoolQ | CSQA | HS | OBQA | PIQA | SIQA | WG |
|---|---|---|---|---|---|---|---|---|---|---|---|
| Baseline | 16.00 | 0.001 | 79.6 | 90.0 | 82.2 | 70.2 | 80.8 | 76.0 | 81.8 | 64.8 | 73.7 |
| Tensor RMS + C | 3.00 | 0.090 | 75.9 | 90.2 | 79.4 | 67.7 | 79.8 | 72.8 | 80.6 | 61.5 | 72.8 |
| Tensor RMS + Sp | 3.05 | 0.129 | 73.9 | 86.5 | 76.5 | 66.3 | 78.6 | 70.6 | 79.8 | 61.4 | 71.6 |
| Block Absmax | 3.25 | 0.132 | 64.9 | 84.2 | 79.4 | 63.3 | 78.2 | 65.4 | 80.0 | 59.5 | 72.7 |
| Channel Absmax | 3.00 | 0.152 | 64.5 | 80.0 | 79.7 | 58.8 | 75.5 | 62.2 | 78.2 | 56.5 | 69.3 |
| Tensor RMS | 3.00 | 0.169 | 20.1 | 27.2 | 61.1 | 19.1 | 35.0 | 26.6 | 56.2 | 32.8 | 51.4 |
| Tensor Absmax | 3.00 | 0.286 | 46.8 | 65.4 | 69.8 | 41.1 | 70.4 | 43.4 | 75.7 | 49.9 | 66.5 |

Table 3: Glossary of terms.

| Symbol | Definition |
|---|---|
| $\theta$ | Parameter vector (whole model or block of parameters) |
| $\tilde{\theta}$ | Reconstructed quantised parameters $\tilde{\theta} = \texttt{dequantise}(\texttt{quantise}(\theta))$ |
| $\tilde{\Theta}$ | Set of reconstructed quantised parameters |
| $F$ | Fisher information matrix of the model $p_\theta(y \mid x)$ |
| $b$ | Bit width; the average number of bits to represent a parameter |
| $D_{KL}\left(p_\theta \| p_{\tilde{\theta}}\right)$ | KL divergence between the predictions of reference and quantised models |
| $T_t$ | Tensor parameter indices; $\theta_{T_t}$ are the parameters of tensor $t$ |
| $B$ | Block size; number of scalar elements in a block |
| $Q$ | Set of quantised representations |
| $b_t^\star$ | Bit width of tensor $t$ under the variable bit allocation scheme |
| $\beta_i$ | Bit width of parameter $i$ in a variable-length code |
| $\rho$ | Scaled KL divergence, $\rho := D_{KL} \cdot 2^{2b}$ |
| $R$ | Root mean square (RMS) error divided by tensor RMS; note signal-to-noise ratio SNR $= 1/R^2$ |

## A  FISHER APPROXIMATION TO KL DIVERGENCE

In this section, we explain the three approximations needed to simplify the problem of minimising KL divergence between reference and quantised model predictions in Equation (1) to minimising the weighted per-tensor squared error of Equation (3). These approximations are a **2nd order Taylor expansion**, **Diagonal Fisher** and **Scaled-identity per-tensor Fisher**. We also briefly discuss their applicability.

**2nd order approximation**  First, we derive the 2nd order approximation of KL divergence in Equation (2) from the definition of KL divergence in Equation (1) and Fisher information $F \in \mathbb{R}^{|\theta| \times |\theta|}$,

$$F := \mathbb{E}_{x \sim \mathcal{X}} \left[ \mathbb{E}_{\mathrm{p}_\theta(y|x)} \left[ (\nabla_\theta \log \mathrm{p}_\theta(y \mid x))(\nabla_\theta \log \mathrm{p}_\theta(y \mid x))^\top \right] \right]. \tag{6}$$

We start by performing a Taylor expansion of $\mathrm{D}_{\mathrm{KL}}\left(\mathrm{p}_\theta \| \mathrm{p}_{\tilde\theta}\right)$ around $\tilde\theta = \theta$,

$$\begin{aligned}
\mathrm{D}_{\mathrm{KL}}\left(\mathrm{p}_\theta \| \mathrm{p}_{\tilde\theta}\right) \approx\ & \mathrm{D}_{\mathrm{KL}}\left(\mathrm{p}_\theta \| \mathrm{p}_\theta\right) \\
& + (\tilde\theta - \theta)^\top \left(\nabla_{\tilde\theta}\, \mathrm{D}_{\mathrm{KL}}\left(\mathrm{p}_\theta \| \mathrm{p}_{\tilde\theta}\right)\right)_{\tilde\theta=\theta} \\
& + \frac{1}{2} \cdot (\tilde\theta - \theta)^\top \left(\nabla_{\tilde\theta}^2\, \mathrm{D}_{\mathrm{KL}}\left(\mathrm{p}_\theta \| \mathrm{p}_{\tilde\theta}\right)\right)_{\tilde\theta=\theta} (\tilde\theta - \theta),
\end{aligned}$$

and observe that the first two terms $= 0$, since $\tilde\theta = \theta$ is a minimum. Now we expand the second derivative

$$\begin{aligned}
\nabla_{\tilde\theta}^2 \mathrm{D}_{\mathrm{KL}}\left(\mathrm{p}_\theta \| \mathrm{p}_{\tilde\theta}\right) &= -\mathbb{E}_{x \sim \mathcal{X}} \left[ \mathbb{E}_{\mathrm{p}_\theta(y|x)} \left[ \nabla_{\tilde\theta}^2 \log \mathrm{p}_{\tilde\theta}(y \mid x) \right] \right], \\
&= \mathbb{E}_{x \sim \mathcal{X}} \left[ \mathbb{E}_{\mathrm{p}_\theta(y|x)} \left[ -\frac{\nabla_{\tilde\theta}^2\, \mathrm{p}_{\tilde\theta}(y \mid x)}{\mathrm{p}_{\tilde\theta}(y \mid x)} + \frac{\nabla_{\tilde\theta}\, \mathrm{p}_{\tilde\theta}(y \mid x)(\nabla_{\tilde\theta}\, \mathrm{p}_{\tilde\theta}(y \mid x))^\top}{\mathrm{p}_{\tilde\theta}(y \mid x)^2} \right] \right], \\
&= \mathbb{E}_{x \sim \mathcal{X}} \left[ \sum_y -\nabla_{\tilde\theta}^2\, \mathrm{p}_{\tilde\theta}(y \mid x) \right] + \mathbb{E}_{x \sim \mathcal{X}} \left[ \mathbb{E}_{\mathrm{p}_\theta(y|x)} \left[ \frac{\nabla_{\tilde\theta}\, \mathrm{p}_{\tilde\theta}(y \mid x)(\nabla_{\tilde\theta}\, \mathrm{p}_{\tilde\theta}(y \mid x))^\top}{\mathrm{p}_{\tilde\theta}(y \mid x)^2} \right] \right].
\end{aligned}$$

The first term $= 0$, since the second derivative can be moved outside the sum, and the second term is equal $F$ as defined in Equation (6), if we expand both logarithmic derivatives. Therefore, $\left(\nabla_{\tilde\theta}^2 \mathrm{D}_{\mathrm{KL}}\left(\mathrm{p}_\theta \| \mathrm{p}_{\tilde\theta}\right)\right)_{\tilde\theta=\theta} = F$, and

$$\mathrm{D}_{\mathrm{KL}}\left(\mathrm{p}_\theta \| \mathrm{p}_{\tilde\theta}\right) \approx \frac{1}{2} \cdot (\tilde\theta - \theta)^\top F\, (\tilde\theta - \theta). \tag{2}$$

*Discussion*  This approximation relies on the assumptions of smoothness and small perturbations. The smoothness assumption is reasonable, as deep learning models are typically optimised with gradient-based methods which have similar smoothness requirements. The small-perturbation assumption is harder to justify at very low bit widths, so we might expect this approximation to be more accurate for higher bit-width formats.

**Diagonal Fisher approximation**  As a further simplification, we assume the cross terms are small, $F_{ij} \approx 0 \quad \forall i \neq j$, so we can simplify the approximate KL divergence to

$$\mathrm{D}_{\mathrm{KL}}\left(\mathrm{p}_\theta \| \mathrm{p}_{\tilde\theta}\right) \approx \frac{1}{2} \sum_i F_{ii} \cdot (\tilde\theta_i - \theta_i)^2. \tag{7}$$

*Discussion*  This is a strong assumption, as there is no particular reason to suspect off-diagonal terms are small. For example, we could imagine quantisation with a change of basis. Take $\theta^{\mathrm{new}} := R\,\theta$, where $R \in \mathbb{R}^{|\theta| \times |\theta|}$ is an orthonormal (rotation) matrix. Optimising in this rotated basis should be equivalent to the original optimisation (as there is no change in difficulty), but with the diagonal Fisher approximation, this is not true. However, the diagonal Fisher approximation is commonly applied as computing the full Fisher matrix is intractable, and it is, for example, central to the widely used Adam optimiser (Kingma & Ba, 2015).

**Scaled-identity per-tensor Fisher approximation** Further to the diagonal approximation, for many of our results we assume that the Fisher information is constant within each parameter tensor, i.e., $F_{ii} = \bar{f}_t$, for $i \in T_t$. With this assumption, the approximate KL divergence simplifies further to

$$D_{KL}\left(p_\theta \| p_{\tilde{\theta}}\right) \approx \frac{1}{2} \sum_t \bar{f}_t \cdot \sum_{i \in T_t} (\tilde{\theta}_i - \theta_i)^2. \tag{3}$$

*Discussion* To justify this assumption, we investigated the structure of the diagonal Fisher information in Figure 12, finding that there is similar or greater variation across tensors than within a tensor. This doesn't discount the value of per-element Fisher statistics, but it indicates that the average Fisher information over a tensor is a quantity of interest, and can vary significantly across tensors.

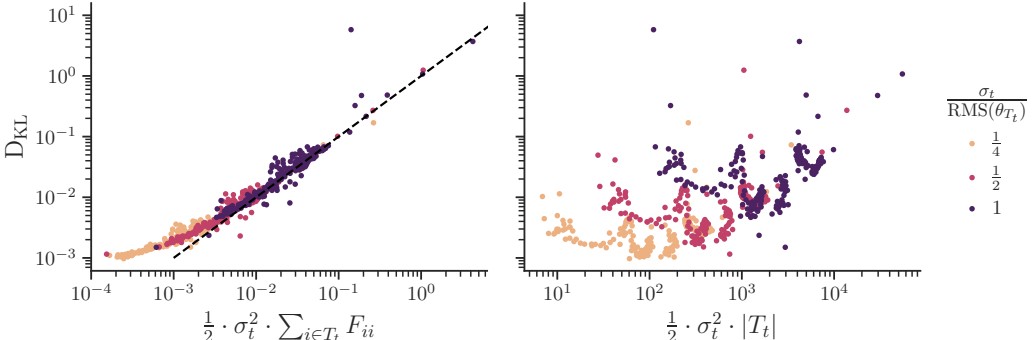

Figure 11: The KL divergence from modifying each parameter tensor in turn with iid noise, compared against *(Left)* the predicted KL divergence from Equation (7), and *(Right)* the scale of perturbation without using Fisher information. For each parameter tensor of Llama 3.1 8B, we perturb $\tilde{\theta}_{T_t} = \theta_{T_t} + \sigma_t \cdot \epsilon$, for a range of $\sigma_t$ and with $\epsilon \sim N^{|T_t|}(0, 1)$, and measure the top-$k$ KL divergence of outputs against the original model. This result indicates that Fisher information is able to predict KL divergence — tensors with higher Fisher information are more sensitive to perturbation.

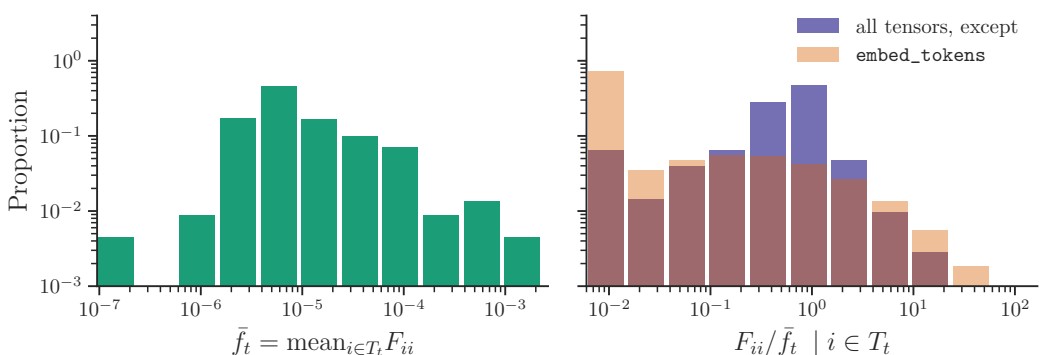

Figure 12: An exploration of the variation in the diagonal of the Fisher information. *(Left)* across different tensors and *(Right)* within each tensor, for Llama 3.1 8B. With the exception of `embed_tokens`, there is a similar level of variation in the Fisher information across tensors as within a tensor. This indicates that the Fisher information may prove useful for both inter and intra -tensor weighted optimisation. Note: the least sensitive tensor is `layers.0.self_attn.q_proj` with mean Fisher of $2.0 \cdot 10^{-7}$, and the most sensitive is `layers.0.self_attn.v_proj` with $1.2 \cdot 10^{-3}$.

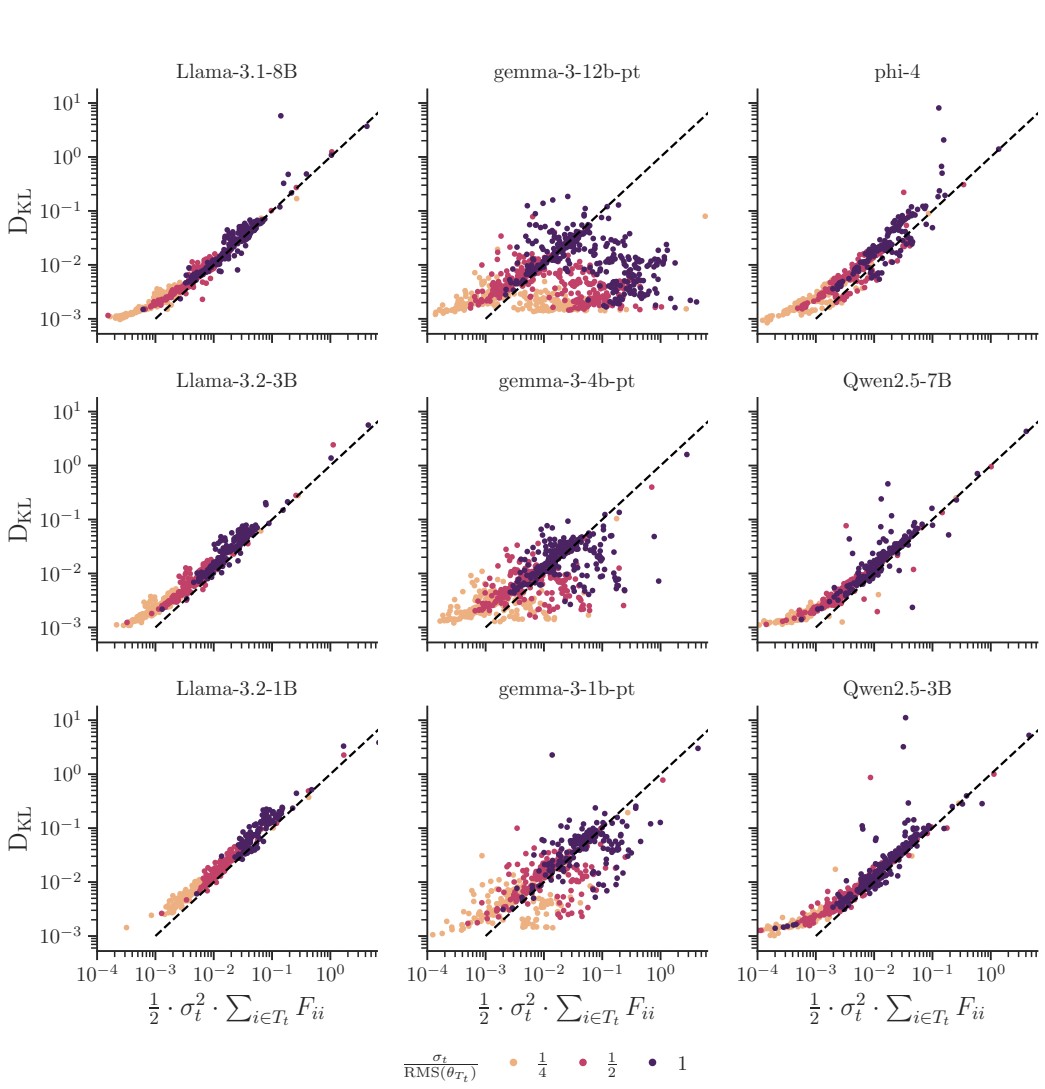

Figure 13: How Fisher information predicts KL divergence given a single-tensor iid random perturbation (as per Figure 11). Most models show a clear trend following the prediction rule, with some outliers. For the Gemma 3 family, there are a set of tensors in early layers for which the Fisher-based prediction overestimates the error. We have found that the flattening of the curve for small noise levels (where the measured KL is higher than predicted) is due to a small bias in top-$k$ KL divergence, compared with true KL divergence.

Table 4: Statistics required for deriving optimal RMS and absmax scaled $\sqrt[3]{p}$ quantisers. See Section B.4 for a derivation of parameters of $\mathcal{D}'$. The expected absmax is taken from extreme value theory (Leadbetter et al., 2012), or in the case of Student-t from our empirical approximation (see Figure 14). Note that $\gamma$ is the Euler–Mascheroni constant.

| Value | Normal ($s$) | Laplace ($s$) | Student-t ($s, \nu$) |
|---|---|---|---|
| RMS $\sqrt{\mathbb{E}\left[\theta_i^2\right]}$ | $s$ | $\sqrt{2} \cdot s$ | $\sqrt{\frac{\nu}{\nu-2}} \cdot s$ |
| $\mathbb{E}\left[\max_{i \in [1..B]} \lvert \theta_i \rvert\right] \approx$ | $\sqrt{2 \log \frac{B}{\pi}} \cdot s$ | $(\gamma + \log B) \cdot s$ | $\left(2 \log \frac{B}{\pi}\right)^{\frac{\nu-3}{2\nu}} \cdot B^{\frac{1}{\nu}} \cdot \sqrt{\frac{\nu}{\nu-2}} \cdot s$ |
| $\mathcal{D}'$ params | $s' = \sqrt{3} \cdot s$ | $s' = 3 \cdot s$ | $\nu' = \frac{\nu-2}{3}$ , $s' = \sqrt{\frac{\nu}{\nu'}} \cdot s$ |

# B  OPTIMAL QUANTISERS

In this section, we present step-by-step recipes for constructing cube root density quantisers (B.1), sketch derivations of the cube root rule (B.2) and uniform density rule (B.3), and derivations for the parameters of $\mathcal{D}'$ (used to apply the cube root rule to Normal, Laplace and Student-t distributions, B.4) and the variable bit allocation scheme (B.5).

## B.1  RECIPES

*Recipe for a $\sqrt[3]{p}$ quantiser*

1. Compute parameters of the target distribution $\mathcal{D}$.
   - For RMS scaling, set RMS $= 1$ and use Table 4 to calculate $s$.
   - For Absmax scaling, set $\mathbb{E}\left[\max_{i \in [1..B]} \lvert \theta_i \rvert\right] = 1$ and use Table 4 to calculate $s$.
2. Compute parameters of $\mathcal{D}'$, which has pdf $p^{\mathcal{D}'} \propto \sqrt[3]{p^{\mathcal{D}}}$ from Table 4.
3. Use the inverse cdf to select quantisation codepoints with density given by $\mathcal{D}'$.

Code examples are given in Section E.

*Recipe for a uniform grid quantiser with compression*

1. Choose a resolution for the grid, $\delta$, so that the quantisation codepoints are $\{\delta \cdot k \mid k \in \mathbb{N}\}$.
2. Either compute the density of values mapped to each codepoint analytically, or via samples.
3. Build an entropy code from this distribution, e.g. using Huffman coding.

To reach a target $b$, this procedure can be wrapped in a search to find an appropriate $\delta$.

## B.2  WITH AN NUMBER OF CODEPOINTS CONSTRAINT — THE CUBE ROOT RULE

The cube root rule states that, under some assumptions, the optimal quantiser for distribution $\mathcal{D}$ should have a codepoint density proportional to the cube root of the pdf of $\mathcal{D}$. This is contrasted with *quantile quantisation* (Gersho & Gray, 1991; Dettmers et al., 2022b), which attempts to distribute quantised values evenly, where the density is proportional to the pdf directly. See Figure 16 for an illustration.

**Derivation**  For a sketch derivation of the cube root rule (Panter & Dite, 1951), consider a piecewise-uniform probability distribution $\{p_i\}$ and a piecewise-uniform quantiser with $n_i$ codepoints in section $i$. Then for a single piece of width $w$, the RMS error is

$$E_i = 2\,n_i \cdot p_i \cdot \int_0^{\frac{w}{2n_i}} \frac{x^2}{w}\,\mathrm{d}x = \frac{p_i \cdot w^2}{12\,n_i^2}.$$

So, with a constraint on number of codepoints i.e., $\sum_i n_i = 2^b$, we use the Lagrange multiplier $\lambda$ to optimise

$$E' = \sum_i \frac{p_i \cdot w^2}{12\,n_i^2} + \lambda \cdot \left(\sum_i n_i - 2^b\right).$$

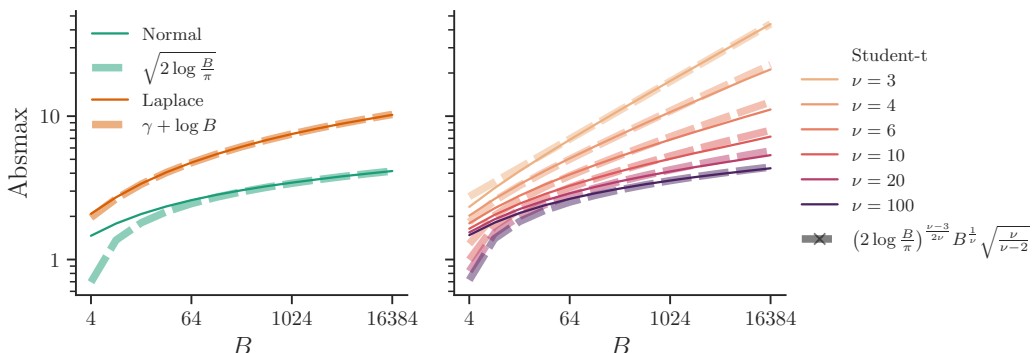

Figure 14: Approximations (*dashed*) to the expected block absmax value for Normal, Laplace and Student-t distributions with scale $s = 1$, versus simulation (*solid*) with $\frac{2^{20}}{B}$ samples. *(Left)* Normal and Laplace distributions. The fit for Normal at small $B \leq 8$ is poor, but typical block sizes are larger than this. *(Right)* Student-t for various degree-of-freedom $\nu \geq 3$, showing good fit across a range of sizes, converging to the Normal approximation as $\nu \to \infty$.

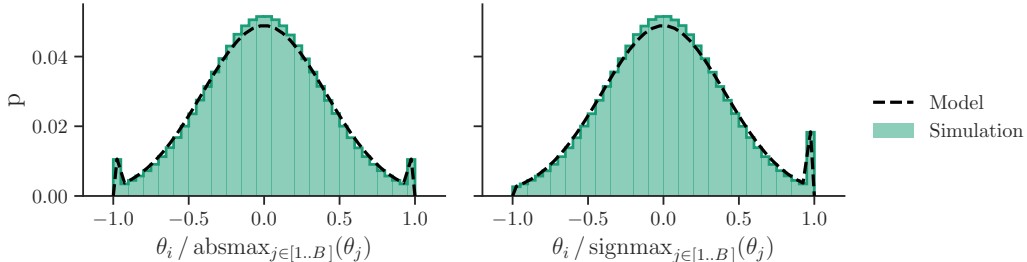

Figure 15: An example of block-scaled Normal data, $B = 64$, using *(left)* absmax and *(right)* signmax scaling: an empirical histogram from sampled data (filled colour) and our mixture model (dashed), using the approximate maximum from Table 4. The empirical marginal distribution is a good fit to our mixture of $\pm 1$ (signmax $+1$) maximum and truncated-Normal non-maxima.

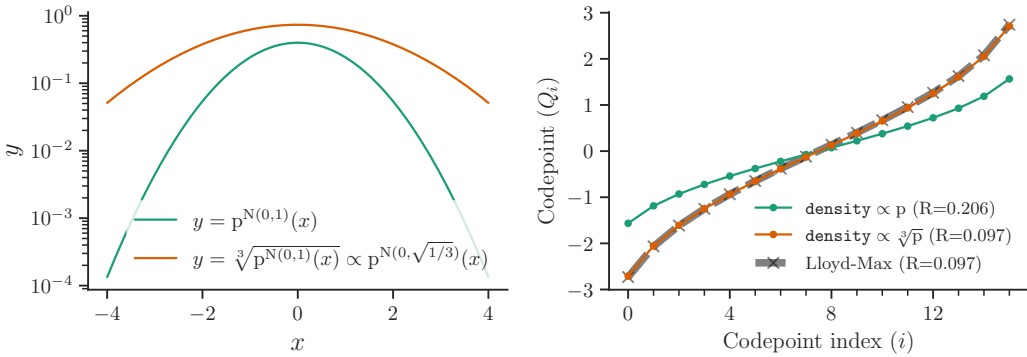

Figure 16: An example of the cube root density rule. *Left:* The density of a standard Normal and $\sqrt[3]{}$ of that density, which is a scaled normal pdf. *Right:* The quantisation curves and error for 4-bit formats derived from the cube root rule, a naive "proportional rule" and a Lloyd-Max quantiser trained on standard Normal samples, showing good match between cube root density and Lloyd-Max.

This gives the gradients

$$\frac{\mathrm{d}E'}{\mathrm{d}n_i} = -\frac{p_i \cdot w^2}{6\,n_i^3} + \lambda,$$

therefore with $\frac{\mathrm{d}E'}{\mathrm{d}n_i}\big|_{n_i=n_i^\star} = 0$, we see that $n_i^\star \propto \sqrt[3]{p_i}$.

### B.3 WITH AN ENTROPY CONSTRAINT — THE UNIFORM DENSITY RULE

The previous method constrained the total number of codepoints, which is appropriate for an un-compressed data stream. If the quantiser is followed by an optimal lossless compressor, we should instead use an entropy constraint:

$$H = -\sum_i n_i \cdot \frac{p_i}{n_i} \log \frac{p_i}{n_i} = b$$

This gives the optimisation objective

$$E'' = \sum_i \frac{p_i \cdot w^2}{12\,n_i^2} + \lambda \cdot \left( \sum_i p_i \log \frac{p_i}{n_i} + b \right),$$

and gradients

$$\frac{\mathrm{d}E''}{\mathrm{d}n_i} = -\frac{p_i \cdot w^2}{6\,n_i^3} - \lambda \cdot \frac{p_i}{n_i},$$

and when $\frac{\mathrm{d}E''}{\mathrm{d}n_i}\big|_{n_i=n_i^\star} = 0$, $p_i$ cancels and $n_i^\star = \mathrm{const}$.

Somewhat surprisingly, the RMS-optimal quantiser when followed by a perfect lossless compressor is a uniform grid (lattice), where the tradeoff between $b$ and $E$ is made by varying the resolution of the grid.

### B.4 DERIVING PARAMETERS OF $\mathcal{D}'$

In this section, we derive the rules for $s'$ and $\nu'$ for the Normal, Laplace and Student-t distributions given in Table 4. For all of these distributions, there is a distribution of the same family, but with different parameters such that the new distribution's pdf is proportional to the cube root of the original pdf.

**Normal**  For a Normal distribution $N(0, s^2)$,

$$\mathrm{p}(x|s) = \frac{1}{\sqrt{2\,\pi \cdot s^2}} \cdot e^{-\frac{x^2}{2\,s^2}}.$$

If we set $\mathrm{p}(x|s') \propto \sqrt[3]{\mathrm{p}(x|s)}$, we see that for some constant $C$

$$\frac{1}{\sqrt[6]{2\,\pi \cdot s^2}} \cdot e^{-\frac{x^2}{6\,s^2}} = \frac{C}{\sqrt{2\,\pi \cdot s'^2}} \cdot e^{-\frac{x^2}{2\,s'^2}},$$

therefore $s' = \sqrt{3}\,s$.

**Laplace**  For a Laplace distribution,

$$\mathrm{p}(x|s) = \frac{1}{2\,s} \cdot e^{-\frac{|x|}{s}}.$$

If we set $\mathrm{p}(x|s') \propto \sqrt[3]{\mathrm{p}(x|s)}$, we see that for some constant $C$

$$\frac{1}{\sqrt[3]{2\,s}} \cdot e^{-\frac{|x|}{3\,s}} = \frac{C}{2\,s'} \cdot e^{-\frac{|x|}{s'}},$$

therefore $s' = 3\,s$.

**Student-t**  For a Student-t distribution,

$$\mathrm{p}(x \mid \nu, s) = \frac{1}{s \cdot \sqrt{\nu} \cdot \mathrm{B}(\frac{1}{2}, \frac{\nu}{2})} \cdot \left( 1 + \frac{x^2}{s^2 \cdot \nu} \right)^{-\frac{\nu+1}{2}}.$$

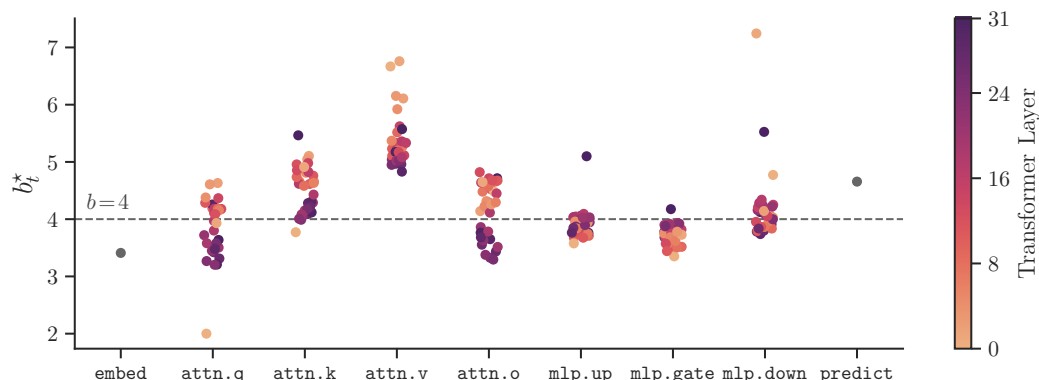

Figure 17: Variable allocation of bits across tensors in Llama 3.1 8B using Equation (5) with a target of 4 bits per parameter. For many element formats, $b_t^\star$ would need to be rounded to the nearest integer. Other members of the Llama 3 and Qwen 2.5 families show a similar trend of requiring additional bits for the attention key and value projections. We suspect this is due to grouped query attention (Ainslie et al., 2023), where the outputs of key and value projections are reused across a group of multiple attention heads.

If we set $p(x \mid \nu', s') \propto \sqrt[3]{p(x \mid \nu, s)}$, we see that for some constant $C$

$$\frac{1}{\sqrt[6]{s^2 \cdot \nu \cdot B(\frac{1}{2}, \frac{\nu}{2})^2}} \cdot \left(1 + \frac{x^2}{s^2 \cdot \nu}\right)^{-\frac{\nu+1}{6}} = \frac{C}{s' \cdot \sqrt{\nu'} \cdot B(\frac{1}{2}, \frac{\nu}{2})} \cdot \left(1 + \frac{x^2}{s'^2 \cdot \nu'}\right)^{-\frac{\nu'+1}{2}},$$

therefore $\nu' = \frac{\nu-2}{3}$ and $s' = \sqrt{\frac{\nu}{\nu'}} \cdot s$.

### B.5 VARIABLE BIT-WIDTH ALLOCATION

In this section, we derive the variable bit-width allocation scheme of Equation (5). We start from the constant-per-tensor Fisher approximation to KL divergence of Equation (3), repeated here:

$$D_{KL}\left(p_\theta \| p_{\tilde\theta}\right) \approx \frac{1}{2} \sum_t \bar{f}_t \cdot \sum_{i \in T_t} (\tilde\theta_i - \theta_i)^2.$$

Now, to forecast how the squared error term depends on bit width, we use the asymptotic limit of Zador (1982), which can be stated as

$$\mathbb{E}\left[(\tilde\theta_i - \theta_i)^2\right] = \epsilon_t^2 \cdot \hat\sigma_t^2 \cdot 2^{-2 \cdot b_t'},$$

where $b_t'$ is the bit width used for tensor $t$, $\epsilon_t$ depends on the distribution of $\theta$ and $\hat\sigma_t^2 := \frac{\sum_{i \in T_t} \mathbb{E}[\theta_i^2]}{N_t} \approx RMS^2(\theta_{T_t})$ with $N_t := |T_t|$. This gives the optimisation

$$\text{minimise} \quad J := \frac{1}{2} \sum_t N_t \cdot \bar{f}_t \cdot \epsilon_t^2 \cdot \hat\sigma_t^2 \cdot 2^{-2 \cdot b_t'},$$

$$\text{subject to} \quad \sum_t b_t' \cdot N_t \le b \cdot \sum_t N_t.$$

Using the Lagrange multiplier $\lambda$, and removing constant factors, we pursue the constrained optimisation,

$$J' = \sum_t N_t \cdot \bar{f}_t \cdot \epsilon_t^2 \cdot \hat\sigma_t^2 \cdot 2^{-2 \cdot b_t'} + \lambda \cdot N_t \cdot (b_t' - b),$$

$$\frac{dJ'}{db_t^\star} = -2 \cdot \ln 2 \cdot N_t \cdot \bar{f}_t \cdot \epsilon_t^2 \cdot \hat\sigma_t^2 \cdot 2^{-2 \cdot b_t^\star} + \lambda \cdot N_t = 0,$$

$$b_t^\star = b^0 + \log_2 \hat\sigma_t + \frac{1}{2} \log_2 \bar{f}_t + \log_2 \epsilon_t,$$

for some constant $b^0$. As a final approximation, we assume that $\epsilon_t = $ const across $t$, so it can be folded into $b^0$ (see Table 5 for justification).

We show an example variable bit allocation computed from this procedure in Figure 17. Most tensors are $\pm 1$ bit from the average, and there is a general trend toward representing some groups of tensors more accurately, e.g. `attn.v`.

Table 5: The variation across tensors of terms that contribute to optimal bit count, for Llama 3.1 8B. Note that $\epsilon$ is estimated based on observed quantisation error $R$, and as such depends on the format used, in this case $b = 4$, Lloyd-Max, Absmax scaling with $B = 64$.

|  | std | $q_{90\%} - q_{10\%}$ |
|---|---|---|
| $\frac{1}{2} \log_2 \bar{f}_t$ | 0.88 | 2.04 |
| $\log_2 \hat{\sigma}_t$ | 0.465 | 1.33 |
| $\log_2 \epsilon_t$ | 0.0302 | 0.0643 |

## C ANALYSIS DETAILS (SIMULATED DATA)

For our analysis on simulated data, we draw data iid from Normal, Laplace or Student-t distributions and measure the quantisation error. Since the scale of a distribution is easily absorbed in the formats we consider, our primary evaluation metric is the ratio of RMS error to data RMS,

$$R := \sqrt{\left(\sum_i [E]_i^2\right) / \left(\sum_i \sum_{j \in [1..B]} \theta_{B \cdot i + j}^2\right)},$$

where $i$ is a block index. We often report $R \cdot 2^b$ for legibility when $b$ varies across an experiment, as $R$ tends to scale as $2^{-b}$.

Unless noted, we sample $|\theta| = 2^{24}$ scalar values for each experiment, and use `float32` compute precision throughout. For compression results, we use a sampling-based method to calculate the model $p^{\mathcal{Q}}$ with a fresh set of samples from the target distribution, and use $+1$ smoothing of the counts (within the training sample range) to avoid zeros.

### C.1 RESULTS

| Question | Figures |
|---|---|
| How to choose between compression & scaling schemes? | 4 |
| How to choose an element format? | 18, 19 |
| How to choose a scale format? | 20, 21 |
| How to choose block size? | 21 |
| Does the cube-root rule work? | 22 |
| Is moment matching sufficient for choosing quantiser scale? | 23 |
| How well does practical compression approach the optimal limit? | 24 |

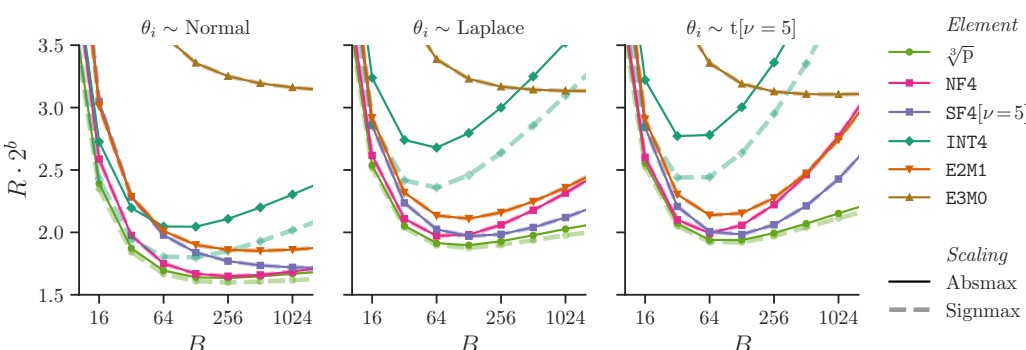

Figure 18: The performance of optimal and extant 4-bit element formats as block size $B$ varies. Note that the total bit width varies with $B$, but is consistent across element formats. We see that the $\sqrt[3]{p}$ formats are marginally better than NF4 and SF4, which don't optimise for RMS error. Of the floating-point and integer formats, E2M1 is generally the best. Signmax quantisation improves INT4 considerably and makes it competitive for Normal data, although performance is still poor for heavier-tailed distributions.

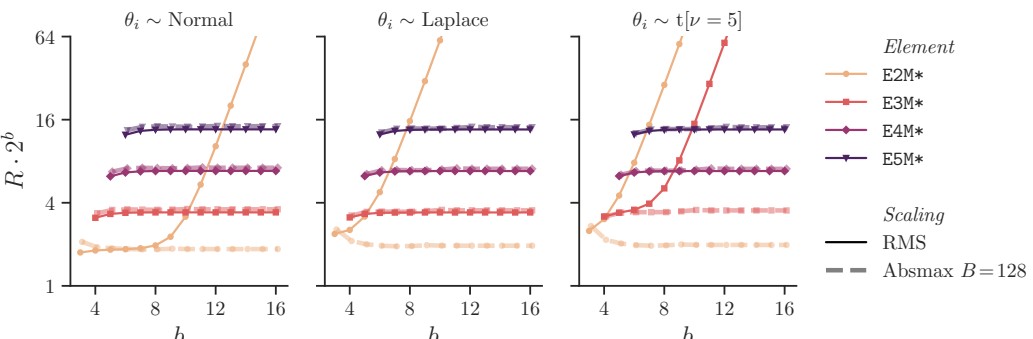

Figure 19: Floating-point element format performance as the total bit width $b$ varies. In general, the optimal number of exponent bits doesn't depend on total bit width. The exception is for RMS scaling, where low-exponent formats eventually stop improving with more mantissa bits (so $R \cdot 2^b$ starts increasing). This is due to the error in quantising the distribution tails, which lie outside the format's range — increasing the number of mantissa bits has negligible effect on range, so this source of error eventually dominates. Note that for this plot, it was important that the bfloat16 scaling factor used round-away rather than round-to-nearest, to avoid range issues from rounding the scale down.

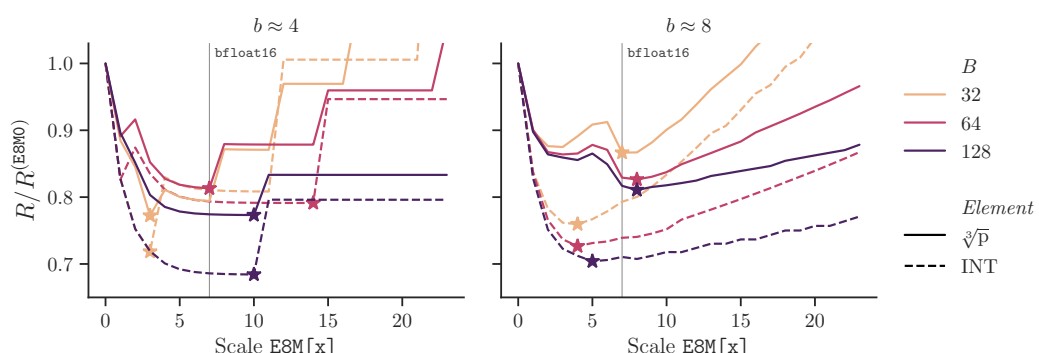

Figure 20: The performance advantage of scale mantissa bits, keeping average bit width $b$ approximately constant by varying the element bit width, for Student-t ($\nu = 5$) data. Both $\sqrt[3]{p}$ and integer formats benefit from 4-10 scale exponent bits, and integers show greater benefit. Note the jumps in the $b \approx 4$ plot are due to a discrete number of codepoints in the element format.

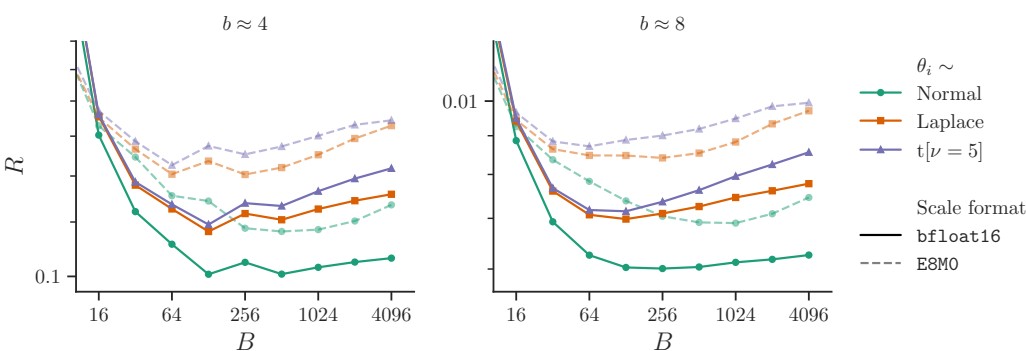

Figure 21: Absmax-scaled format error versus block size $B$, for different approximate bit widths, data distributions and scale format. As $B$ decreases, the element bit width is reduced to keep $b = b^{\text{element}} + \frac{b^{\text{scale}}}{B}$ approximately constant. `bfloat16` (or E8M7) outperforms the mantissa-less E8M0 format. The optimum for Normal data is generally slightly to the right of that for heavy-tailed Laplace and Student-t distributions, generally in the range 64–256.

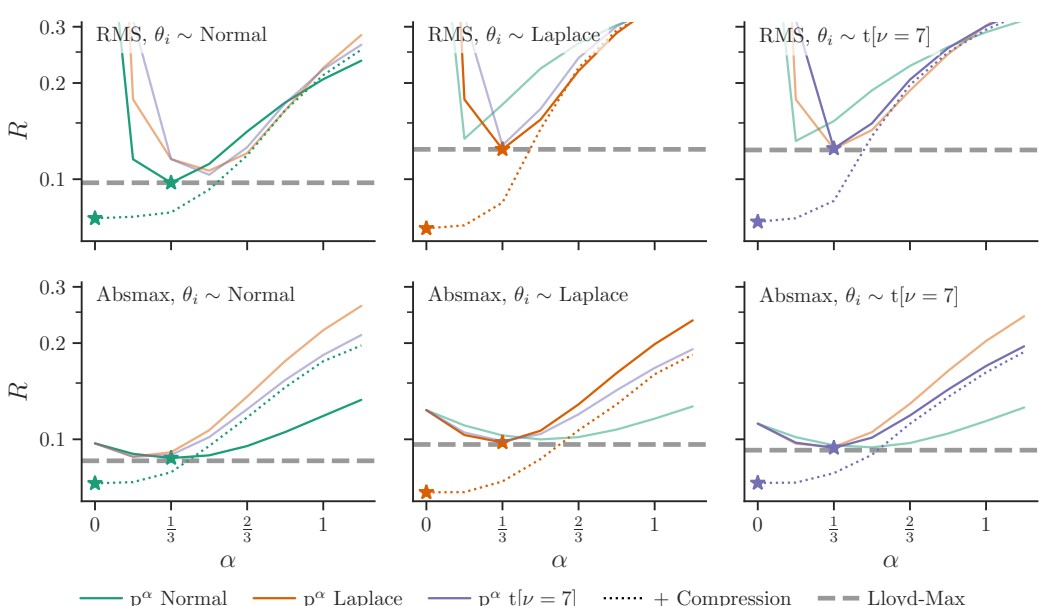

Figure 22: Validation of optimal 4-bit $\sqrt[3]{\mathrm{p}}$ quantisers via simulation. We generalise the $\sqrt[3]{\mathrm{p}}$ rule to a $\mathrm{p}^\alpha$ rule for various $\alpha$ (horizontal axis) and try quantisers derived from different distributions (hue) using moment-matching. *(Top)* RMS scaling. *(Bottom)* Absmax scaling, $B = 64$. We find that the best quantiser is consistently the matching $\sqrt[3]{\mathrm{p}}$ ($\alpha = \frac{1}{3}$), which performs comparably to a Lloyd-Max trained quantiser. We also show the curve for a compressed quantiser with $b \approx 4$, which has optimum at $\alpha = 0$, i.e. a uniform grid that is independent of the pdf.

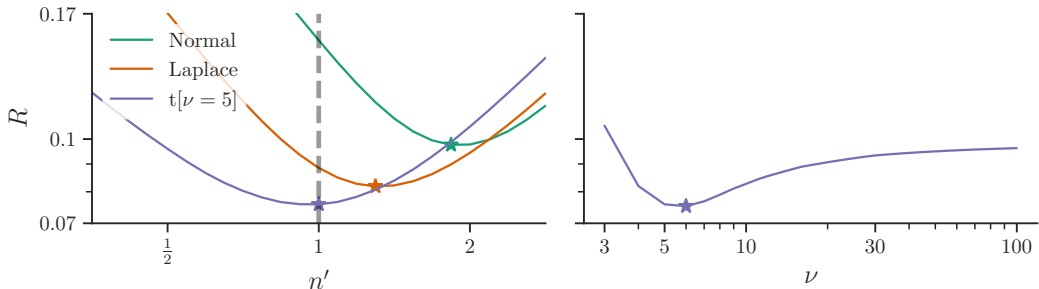

Figure 23: Search to find the best 5-bit quantiser parameters after RMS scaling of data generated from a Student-t ($\nu = 5$) distribution. *(Left)* search over the scale applied to the quantiser, such that $\tilde{\theta}_i = n' \cdot \mathtt{dequantise}\left(\mathtt{quantise}\left(\frac{\theta_i}{n'}\right)\right)$. Note that each quantiser (Normal, Laplace, etc) is optimal for data of their matching distribution, with RMS $= 1$. For the correct Student-t quantiser, RMS moment matching ($n' = 1$) works well, but moment matching performance is suboptimal for mismatched quantisers. *(Right)* search to find the correct Student-t quantiser $\nu$. For each $\nu$, we search for the scale $n'$ that minimises $R$.

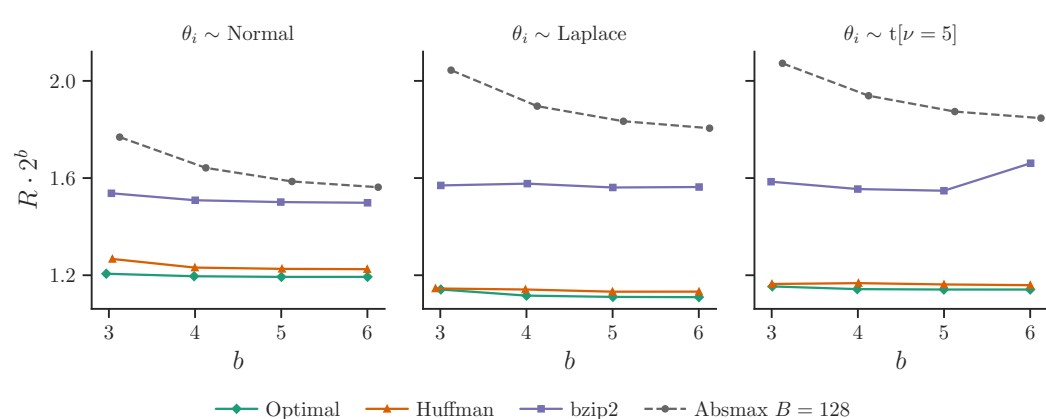

Figure 24: The performance of practical compressors using RMS scaling and $\sqrt[3]{\text{p}}$ element formats, compared with the theoretical limit, over $|\theta| = 2^{20}$ samples. An elementwise Huffman code (Huffman, 1952) using `dahuffman` (Lippens, 2017) performs close to optimal. Bzip2 doesn't reach the same compression ratio, however it still outperforms an uncompressed block format.

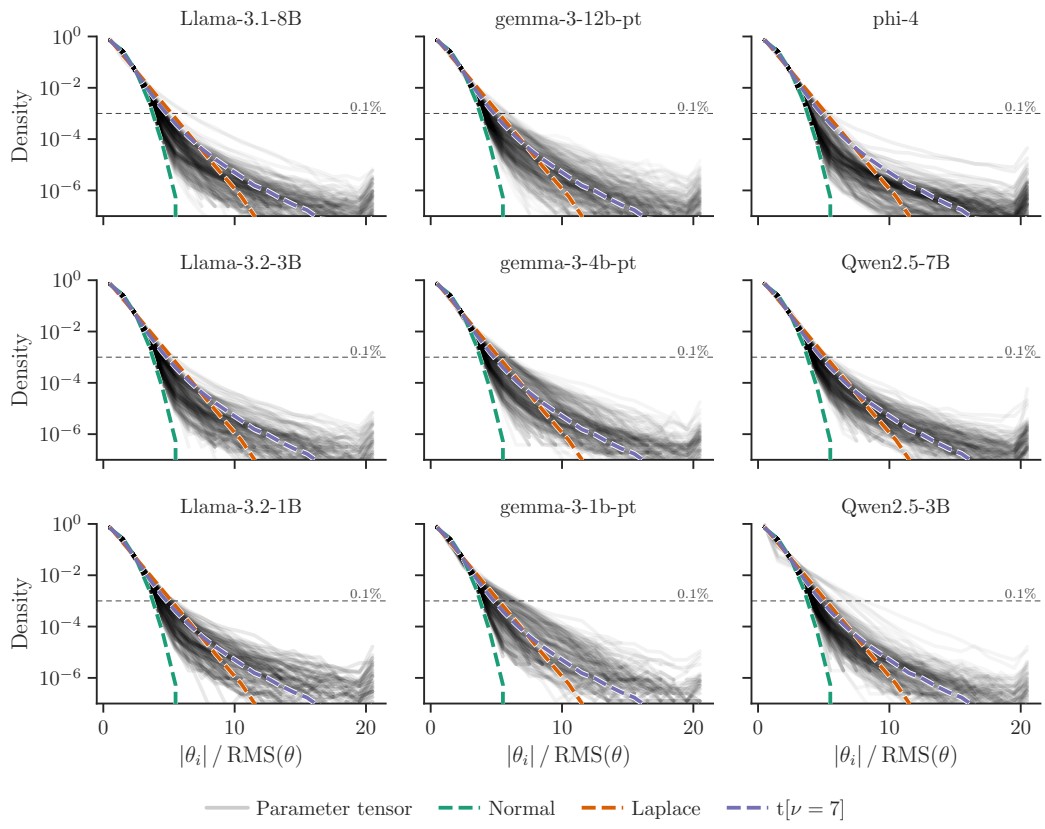

Figure 25: A histogram of absolute parameter values for various models. Each line corresponds to a parameter tensor in the model. As we care about tails not scales (the overall scale of a tensor is easily absorbed into a scaling factor), we divide each parameter value by the RMS of the tensor. We note that different models show the same general trends: heavy tails that seem closest to a Student-t distribution in shape, with some variability across tensors in the model.

## D    EXPERIMENTAL DETAILS

Our language modelling experiments use the WikiText-103 (Merity et al., 2017) combined validation and test sets. For each sequence in the dataset, we generate full sequence (teacher-forcing) logits from the reference model and test model for evaluation using cross entropy and top-$k$ KL divergence, which is described below. Hyperparameters are given in Table 6. The 11 models evaluated are: Llama 3.1 8B, Llama 3.2 1B, Llama 3.2 3B, Phi 4 (14B), Qwen 2.5 {0.5B, 1.5B, 3B, 7B} and Gemma 3 {1B, 4B, 12B} (Dubey et al., 2024; Yang et al., 2024; Kamath et al., 2025; Abdin et al., 2024). Where multiple variants exist, we use the bare pretrained model.

The division of parameters into tensors follows the Huggingface `transformers` (Wolf et al., 2020) checkpoints, which differ slightly between models. For example, Phi-4 contains a single "stacked" projection matrix for query-key-value, while the other models tested store them separately.

Our $k$-means results use a custom implementation which iterates until the proportion of cluster assignments that change drops below $10^{-4}$ and uses `k-means++` (Arthur & Vassilvitskii, 2007) initialisation for RMS-scaled data and uniform $(-1, 1)$ initialisation for absmax-scaled data — settings which we found to be robust during early testing.

Table 6: Experimental settings.

| Hyperparameter | Value |
|---|---|
| Eval Sequence length | 4096 |
| Eval KL top-$k$ | 128 |
| Eval tokens | $\approx 5 \cdot 10^5$ |
| Fisher estimation tokens | $4 \cdot 10^6$ |
| Reference parameter format | `bfloat16` |
| `transformers` version | `4.51.3` |
| Scale search range | $[2^{-2}, 2^{-1.75}, \ldots, 2^2]$ (17 steps) |
| Student-t $\nu$ search range | `logspace`$(\log_2 3, \log_2 100, \text{steps}=12, \text{base}=2)$ |
| *QAT* | |
| Batch $\cdot$ Sequence length | $64 \cdot 1024$ |
| Steps | 8192 |
| Optimiser | Adam $\beta_{1,2} = (0.9, 0.95)$ |
| LR | Cosine, $\eta = 2^{-14-b^{\text{elem}}}$ |

**Top-$k$ KL divergence**    Our comparison metric is top-$k$ KL divergence, defined for a single pair of logits that specify $\mathrm{p}_\theta(y_i \mid x)$ and $\mathrm{p}_{\tilde{\theta}}(y_i \mid x)$ as

$$\mathrm{D_{KL}}^{\text{top-}k}(\mathrm{p}_\theta, \mathrm{p}_{\tilde{\theta}}) := \left( \sum_{y \in \mathrm{argtop}k(p)} p_y \cdot \log \frac{p_y}{q_y} \right) + p^{\text{tail}} \cdot \log \frac{p^{\text{tail}}}{q^{\text{tail}}}$$

$$\text{where } p_y := \mathrm{p}_\theta(y \mid x) \text{ and } q_y := \mathrm{p}_{\tilde{\theta}}(y \mid x) ,$$

$$p^{\text{tail}} := \sum_{y \notin \mathrm{argtop}k(p)} p_y ,$$

$$q^{\text{tail}} := \sum_{y \notin \mathrm{argtop}k(p)} q_y .$$

Note that the top-$k$ always applies to the reference model, never the target model. The *tail* term is required to ensure that the KL divergence is $\geq 0$. The logic is equivalent to creating a modified distribution where the non-top-$k$ classes are collapsed into a single output class, followed by regular KL divergence over $k + 1$ classes.

We use top-$k$ KL divergence rather than full KL divergence because the vocabulary size of language models (typically $> 10^5$) makes it prohibitive to store a dataset of reference logits. Top-$k$ KL

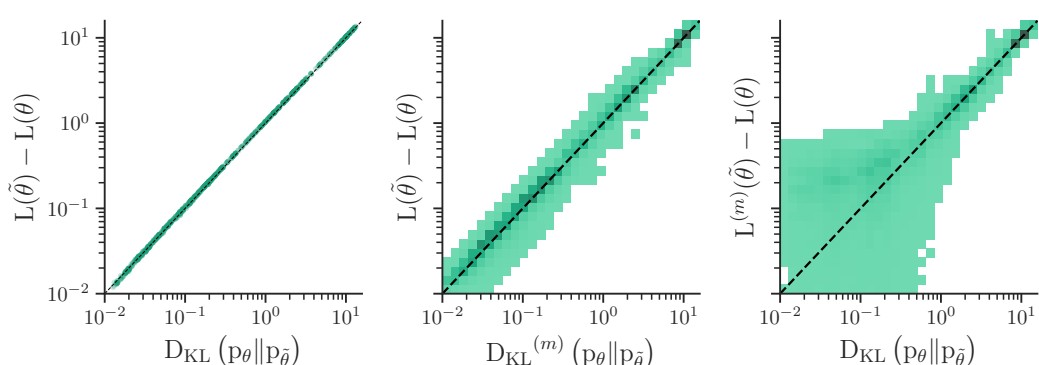

Figure 26: The correlation of top-$k$ KL divergence with change in cross entropy, $L(\tilde{\theta}) - L(\theta)$, for a wide sweep of runs on Llama 3.1 8B. *(Left)* mean KL versus change in cross entropy where each point is a quantisation experiment, showing excellent agreement. *(Center)* histogram of sample KL (individual sequence results) versus mean change in cross entropy for the run, and *(Right)* histogram of mean KL for the run versus change in sample cross entropy. These show that top-$k$ KL divergence provides a much tighter error estimate than cross entropy when the degradation is small, however it is likely that a per-sequence version of the cross entropy increase (i.e. $L^{(m)}(\tilde{\theta}) - L^{(m)}(\theta)$) would give a similar benefit.

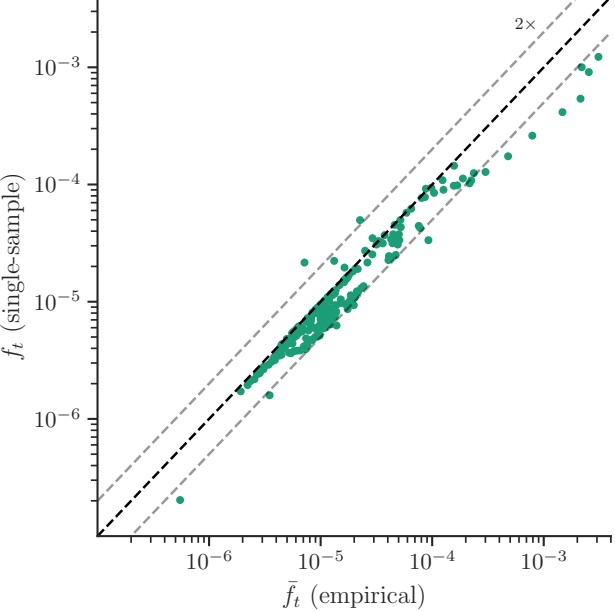

Figure 27: A comparison of the estimated Fisher from Equation (8), which uses a sampled token from the model versus empirical Fisher using the target from the dataset, for each tensor in Llama 3.1 8B. The empirical Fisher is generally a little larger, due to a mismatch between the model's predictive distribution and the data distribution.

divergence stores $2 \cdot k$ scalar values per token, an index and a log-probability, versus a log-probability per vocabulary term for full KL divergence.

In Figure 26 we investigate the benefit of using KL divergence over cross entropy. Since the two metrics are extremely well correlated, we would expect no difference in the outcomes. However the KL divergence gives tighter error bounds on the estimate, since the quantised and original models are compared for each token rather than once for the aggregate statistic.

**Fisher estimation** To estimate the diagonal of the Fisher information defined in Equation (6) for the reference model, we sample text sequences from the WikiText-103 training set. For each sequence, we generate logits from the model, sample a single output token per position in the sequence from the predicted distribution and backpropagate the cross entropy loss of the sampled token to get the gradient with respect to activations. We replace the calculation of parameter gradients with a custom version, which squares the gradients before accumulation (see Section E.3). That is, we calculate

$$F_{ii} \approx \frac{1}{M \cdot L} \sum_{m \in [1..M]} \sum_{p \in [1..L]} \left( \nabla_{\theta_i} \log p_\theta(\hat{y}_p^{(m)} \mid x_{<p}^{(m)}) \right)^2 \quad \text{where } \hat{y}_p^{(m)} \sim p_\theta(y \mid x_{<p}^{(m)}). \quad (8)$$

over $M = 1024$ sequences of length $L = 4096$. Note that we use a sampled target label rather than the ground truth from WikiText in order to be closer to estimating the Fisher rather than empirical Fisher, a difference explored by Kunstner et al. (2019), at the cost of increased variance of our estimator. We explore the correlation between these in Figure 27. Despite this effort, since we use "teacher forcing" of inputs in an autoregressive setting, the method remains somewhat empirical.

Since this method accumulates the diagonal Fisher, it stores $|\theta|$ additional values, a similar amount of memory to training with SGD. Although the parameters may be represented in `bfloat16`, it is important to accumulate the Fisher statistics in a format with more mantissa bits, as `bfloat16` updates will be swamped after $\mathrm{O}(2^8)$ steps. To support Fisher estimation with limited accelerator memory, we implement a 2-stage accumulator that accumulates $64$ steps in `bfloat16` on device, then accumulates these batched updates in `float32` on the host CPU.

**Moment matching baselines** For RMS scaling with $\sqrt[3]{p}$ formats, the moment matching baseline sets the RMS of the quantiser to match that of the data. For standard formats it scales such that data RMS $= 1$ in the case of E2M* and $\frac{2^{b-1}-1}{\sqrt{3}}$ (to match the RMS of a uniform distribution) in the case of INT. With Absmax scaling, the moment matching baseline sets the scale such that the minimum of the positive and negative range of the quantiser matches that of the normalised data, i.e. to cover $(-1, 1)$.

**Quantisation aware training** We initialise two copies of the pretrained checkpoint of a given model: one to serve as a reference model to produce target logits, and one for quantising using QAT. We replace each parameter in the quantised model with a compute graph which performs the following:

1. Calculate block, channel or tensor *scale* from the *master* parameter tensor.
2. Divide the master by the scale.
3. Round to nearest quantisation centroid, with a straight-through estimator (identity gradient operator) in the backwards pass.
4. Multiply by the scale.
5. (If applicable), replace parameters at *sparse indices* with *sparse values*.

Note that quantisation centroids are computed when converting the model, based on the format under test, and are not updated during training. Only master parameters and sparse values are updated during training, as the scale is calculated based on the master parameter, using absmax or RMS as appropriate.

After conversion, we train the quantised model on batches sampled from SlimPajama (Soboleva et al., 2023), with a *full* KL divergence loss against the reference model output. Early learning rate sweeps indicated that the best learning rate depended on the target bit width $b$, and that $\eta \propto 2^{-b}$ was a reasonable heuristic. Key training hyperparameters are given in Table 6.

**Downstream tasks** We use OLMES (Gu et al., 2025) for downstream evaluation over the following tasks: ARC Challenge "ARC-c" (MC), ARC Easy "ARC-e" (MC), BoolQ (Cloze), CSQA (MC),

HellaSwag "HS" (Cloze, limit to 1000 examples), PIQA (Cloze), SocialIQA "SIQA" (MC) and Winogrande "WG" (Cloze). Note: MC = Multiple Choice.

Our summary metric *downstream mean accuracy ratio* is computed by taking the ratio of downstream task accuracy to the unquantised baseline accuracy, then clipping to the range $[0, 1]$, before averaging across tasks.

## D.1 RESULTS

| Question | Figures |
|---|---|
| How to choose between (compression, scaling, outlier) schemes? | 1, 8, 28 |
| Does Downstream/QAT change this? | 7, 9; Tables 1, 2 |
| How do random rotations help? | 29 |
| Does variable bit allocation help? | 6, 30 |
| How to choose an element format | 31, 32 |
| How to choose a scale format? | 33 |
| How to choose block size? | 33 |
| Signmax, Asymmetric or Symmetric scaling? | 34 |
| Moment matching or scale search? | 35 |

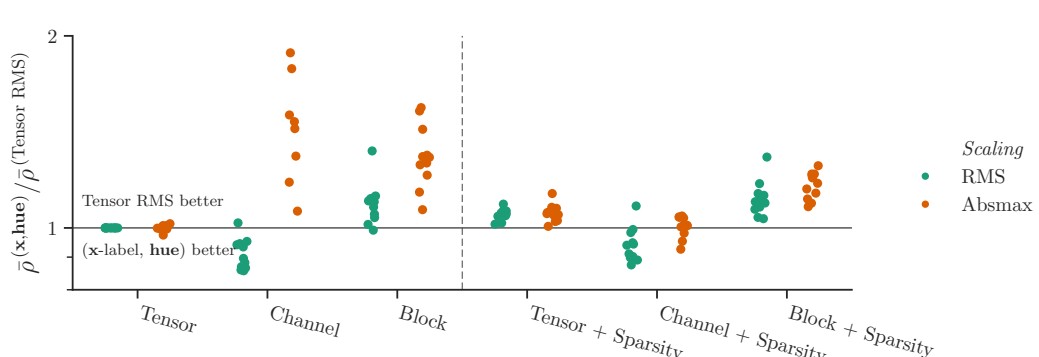

Figure 28: The average change in scaled KL for various scaling schemes and sparsity, *when combined with optimal lossless compression*. Note that each point is the scaled KL for a model, averaged over bit widths, and divided by the tensor RMS baseline. In the presence of lossless compression, there is no benefit to block scaling or separating sparse outliers, consistent with our claim that they exploit the same variable-length encoding benefit offered by compression. The only scaling mode that outperforms simple tensor RMS scaling when combined with compression is channel RMS scaling, which exploits structure in the tensor data.

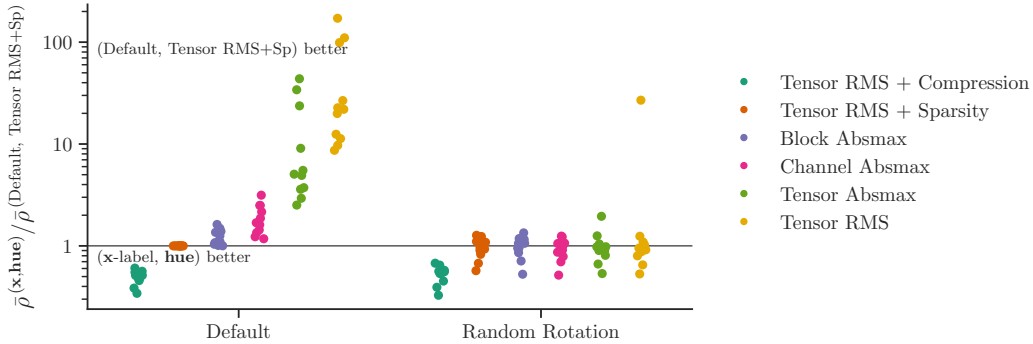

Figure 29: An evaluation of random rotations, where the rotations $V$ and $W$ are applied before quantising the rows and columns respectively of a 2D parameter tensor, i.e. $\tilde{\theta} = V^\top \texttt{dequantise}(\texttt{quantise}(V\,\theta\,W))W^\top$. Since we expect rotated parameters to be roughly normally distributed, we use the $\sqrt[3]{p}$ normal quantiser, optionally with a block scaling scheme, sparse outliers or compression. Our results show that random rotations are useful for fixed-length schemes such as tensor scaling without sparse outliers, but unnecessary for schemes that employ variable-length coding. This is what we'd expect: rotations transform heavy-tailed marginal distributions, where fixed-length quantisation performs much worse than variable-length quantisation (Figure 4 *(right)*), towards the Normal distribution, for which fixed-length quantisation performs better (Figure 4 *(left)*).

Note that the outlier point for Tensor RMS scaling with rotation corresponds to the Phi-4 model, which is likely an experimental issue — for sake of memory, we skip rotations where the dimension is too large (e.g. embedding vocabulary dimension), and with the large hidden size of Phi-4, our code also skipped rotating the output dimension of the stacked MLP up-and-gate projection.

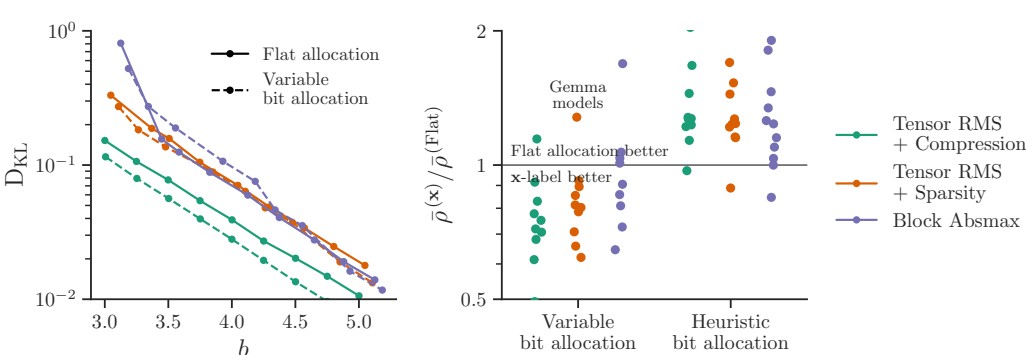

Figure 30: The performance of the Fisher-based variable bit allocation scheme of Equation (5) for `codeparrot/github-code` (Tunstall et al., 2022), when the Fisher information was calculated over WikiText, a substantially different dataset. *(Left)* the tradeoff curve for Llama 3.1 8B, showing a general shift to the left, although some settings for absmax scaling are degraded. *(Right)* the average scaled KL of different bit allocation schemes compared against flat allocation, for all models. Much of the in-domain improvement is retained, indicating that the Fisher information can generalise across datasets. Note that the *heuristic bit allocation* scheme allocates $+2$ bits to all parameters in the first 2 and last 2 transformer layers, and to embedding and final projection parameters; this performs poorly.

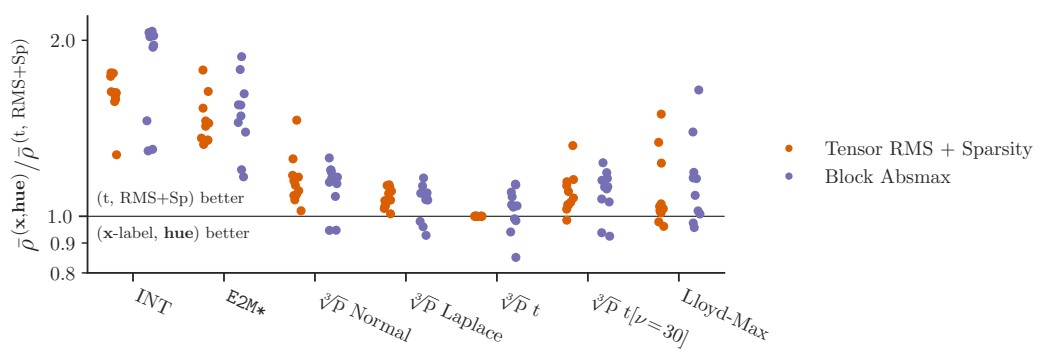

Figure 31: A comparison of different element formats, each point the best setting for a given model, over {moment matching/search/Fisher-weighted search, symmetric/asymmetric variant}, compared with Student-t with RMS scaling and sparse outliers. No setting consistently beats this baseline across models; surprisingly, this includes Lloyd-Max with Fisher weighting.

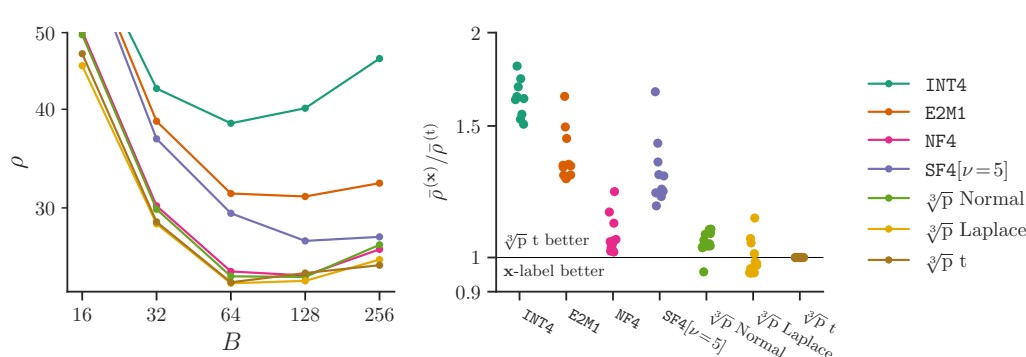

Figure 32: A comparison of $\sqrt[3]{p}$ against extant formats with block absmax scaling, 4-bit elements and `bfloat16` scale, i.e. $b = 4 + \frac{16}{B}$. *(Left)* Llama 3.1 8B performance as $B$ varies. We see that the $\sqrt[3]{p}$ formats and `NF4` perform similarly. Note that $\sqrt[3]{p}$ Normal is different from `NF4`, since $\sqrt[3]{p}$ formats optimise for RMS not incompressibility and use a model of the block-maximum, meaning that the curve depends on $B$. *(Right)* average performance across different models, where each point gives the average $\rho$ across block size, divided by the performance of (model, $\sqrt[3]{p}$ Student-t). We see that $\sqrt[3]{p}$ Laplace and Student-t perform best in general, and there is little to choose between $\sqrt[3]{p}$ Normal and `NF4`. Surprisingly, `SF4` is worse, at odds with the findings of Dotzel et al. (2024). One possible explanation for the difference is our use of a `bfloat16` scale, which provides a tighter bound on the block maximum, compared with an `E8M0` exponent.

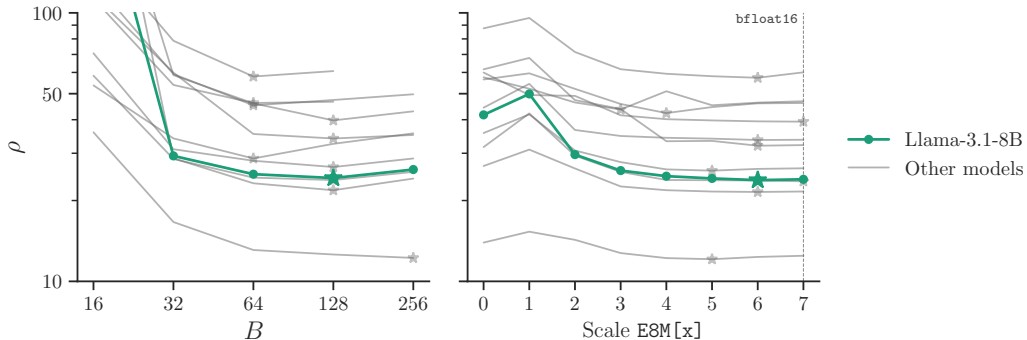

Figure 33: Hyperparameter sweep for block absmax formats using the Student-t element quantiser and $b \approx 4$. *(Left)* block size sweep, showing that almost all models agree on $B = 128$, given a `bfloat16` scale, consistent with our simulations in Figure 21. *(Right)* scale mantissa bits sweep with round-away, showing that most models benefit from 4-6 scale mantissa bits, given $B = 128$, consistent with Figure 20. For both sweeps, a fair comparison is made by adjusting the element width to account for the different scale overhead. For example, for $B = 64$ with a `bfloat16` scale, the element bit width is set as close to $4 - \frac{16}{64}$ as possible.

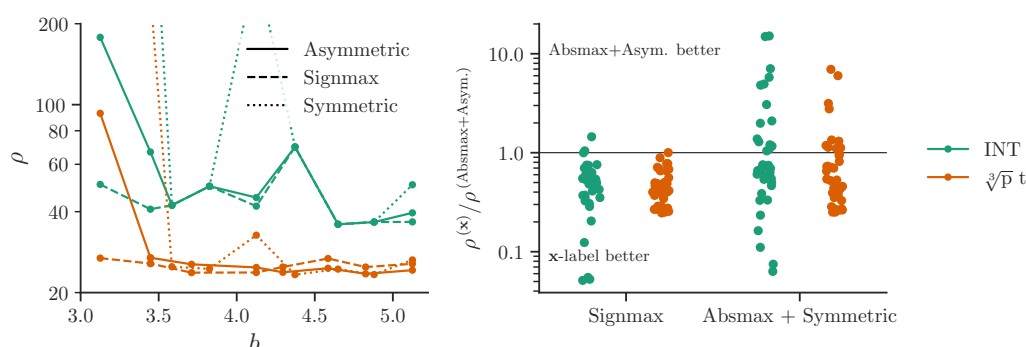

Figure 34: A comparison of scaling variants (see Figure 3) for INT and $\sqrt[3]{p}$ Student-t element formats, using block scaling with $B = 128$. *(Left)* the tradeoff curve for Llama 3.1 8B, showing that signmax outperforms regular absmax scaling at small $b$. Symmetric scaling, which does not include a representation for $0$ does not perform consistently for this model. *(Right)* scaled KL over all models, relative to the absmax + asymmetric variant. The improvement from signmax is consistent. The symmetric format is sometimes better and sometimes worse than asymmetric.

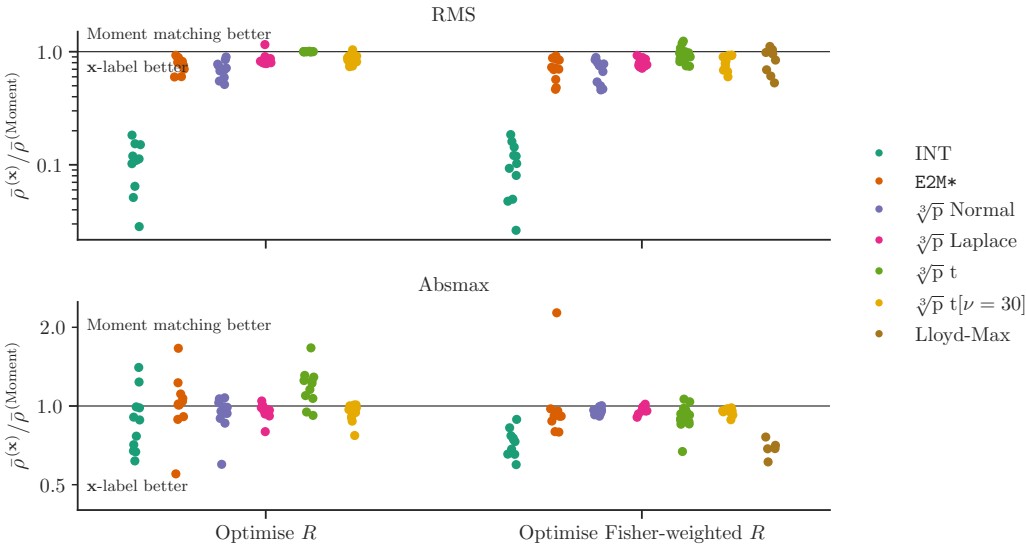

Figure 35: Moment matching vs search over quantiser scale to optimise $R$ and Fisher-weighted search. Each point corresponds to the average $\rho$ over bit width for one of the 11 models tested. The results suggest that search is helpful for RMS scaling, but not reliable for absmax scaling, although Fisher weighting seems to help here. Note that all formats for Qwen2.5-3B perform very badly using Fisher-weighted search, with a ratio $\frac{\rho^{\text{(Fisher)}}}{\rho^{\text{(Moment)}}} > 2$. Only one of these results is visible in-range.

# E CODE EXAMPLES

This section provides illustrative implementations that compute optimal quantisation curves for RMS and block absmax Normal, Laplace and Student-t distributions as well as code to estimate the diagonal of the Fisher information matrix. For the full implementation used for our experiments, please see *(see supplementary materials)*.

## E.1 CUBE ROOT DENSITY (RMS SCALING)

Illustrative implementations of 4-bit cube root density quantisation of different distributions (symmetric variant).

**Normal**

```
b = 4
p = torch.linspace(0, 1, 2**b + 2)
Q = torch.tensor(scipy.stats.norm.ppf(p[1:-1], scale=sqrt(3)))

def quantise(x): return torch.bucketize(x, (Q[1:] + Q[:-1]) / 2)
def dequantise(i): return Q[i]
```

Note the `scale` for the ppf (inverse cdf) is set to $\sqrt{3}$, according to the rule from Table 4.

**Laplace** (RMS $= 1$):

```
b = 4
p = torch.linspace(0, 1, 2**b + 2)
Q = torch.tensor(scipy.stats.laplace.ppf(p[1:-1], scale=3/sqrt(2)))
```

**Student-t** ($\nu = $ df, RMS $= 1$):

```
b, df = 4, 7
p = torch.linspace(0, 1, 2**b + 2)
Q = torch.tensor(scipy.stats.t.ppf(p[1:-1], (df-2)/3, scale=sqrt(3)))
```

## E.2 CUBE ROOT DENSITY (BLOCK ABSMAX SCALING)

Illustrative implementations of 4-bit cube root density quantisation of different distributions, scaled by their block absmax (symmetric variant).

**Normal**

```
b, block_size = 4, 64
p = torch.linspace(0, 1, 2**b)
scale = sqrt(3 / (2 * log(block_size/pi)))
Q = torch.tensor(scipy.stats.truncnorm.ppf(p, -1/scale, 1/scale, scale=scale))
```

Note the `scale` for the inverse cdf is $\frac{s'}{\mathbb{E}[\max_i \theta_i]}$ from Table 4.

**Laplace**

```
def trunclaplace_ppf(q, x0, x1, scale):
    c0, c1 = scipy.stats.laplace.cdf([x0*scale, x1*scale], scale=scale)
    return scipy.stats.laplace.ppf(c0 + (c1-c0)*q, scale=scale)

b, block_size = 4, 64
p = torch.linspace(0, 1, 2**b)
scale = 3 / (0.57721566 + log(block_size))
Q = torch.tensor(trunclaplace_ppf(p, -1/scale, 1/scale, scale=scale))
```

**Student-t** ($\nu = $ df)

```
def trunct_ppf(q, df, x0, x1, scale):
    c0, c1 = scipy.stats.t.cdf([x0*scale, x1*scale], df, scale=scale)
```

```
        return scipy.stats.t.ppf(c0 + (c1-c0)*q, df, scale=scale)

b, block_size, df = 4, 64, 7
p = torch.linspace(0, 1, 2**b)
scale = (2*log(block_size/pi))**((3-df)/(2*df)) * block_size**(-1/df) * sqrt(3)
Q = torch.tensor(trunct_ppf(p, (df-2)/3, -1/scale, 1/scale, scale=scale))
```

### E.3 FISHER ESTIMATION

Illustrative code for wrapping a `torch.nn.Linear` layer with logic to compute the sum of squared gradients in order to estimate the diagonal of the Fisher information matrix.

```python
class FisherWrappedLinear(torch.nn.Module):
    def __init__(self, m: torch.nn.Linear):
        super().__init__()
        self.m = m
        self.gW2 = torch.zeros_like(self.m.weight, dtype=torch.float32)

    def forward(self, x):
        y = self.m(x)
        y.requires_grad_(True).register_hook(
            lambda gy: self.gW2.addmm_(
                gy.detach().flatten(0, -2).float().square().T,
                x.detach().flatten(0, -2).float().square(),
            ) is None or None
        )
        return y
```

