# OpenReview forum: "Optimal Formats for Weight Quantisation"
_ICLR.cc/2026/Conference — Submitted to ICLR 2026_

### Official Review · Reviewer_VQ1o · 2025-10-25

**Soundness:** 3
**Presentation:** 2
**Contribution:** 2
**Rating:** 4
**Confidence:** 4

**Summary:**

The paper explores various tweaks to scalar quantization formats and proposes minor improvements over existing practices.
However, I am not sure what is the main takeaway here.

**Strengths:**

The paper does a thorough evaluation of various settings of scalar quantization.

**Weaknesses:**

- The overall message of the paper is unclear.
- Concepts are sometimes not introduced or introduced in inappropriate places (Lloyd-max, Huffman coding).
- Some proposed schemes (i.e. huffman compression) are impractical and cannot be used in efficient GPU kernels.
- Saying that block schemes perform variable-length encoding seems far fetched.
- Variable bit allocation could be explored more and compared to previous works like Evopress.

**Questions:**

What is the main takeaway for people who work with scalar quantization here?

---

> ### Author Response · Authors · 2025-11-20
>
> Thank you for your review and questions. We hope we are able to address your concerns below.
>
> ---
>
> > What is the main takeaway for people who work with scalar quantization here?
>
> Our main takeaway is that compression schemes that effectively perform variable-length encoding achieve optimal performance under a fixed memory budget. As analysed in the paper, there are various ways of achieving this: directly, by using entropy coding, or indirectly, by using block-scaled formats such as MXFP, or formats with sparse outlier storage. Practitioners can therefore view these methods as points along a unified design space for achieving optimal weight compression.
>
> > Saying that block schemes perform variable-length encoding seems far fetched.
>
> Our claim is based on the observation (Figure 4) that block-absmax formats can outperform _optimal_ fixed-length codes. From the perspective of an individual parameter, they must then perform variable-length coding. Alternatively, block schemes can be viewed as a form of fixed-length _vector_ codes. For vectors of sufficient size, fixed-length vector codes can perform as well (or even better) than variable-length scalar codes. The scalar perspective nevertheless remains valid (as explored in Figure 5), since the block scale represents the extreme value within the block, effectively allocating more bits to that value.
>
> > Concepts are sometimes not introduced or introduced in inappropriate places (Lloyd-max, Huffman coding).
>
> Thank you for this helpful comment – we have updated the paper to properly introduce these terms on first use.
>
> > Some proposed schemes (i.e. huffman compression) are impractical and cannot be used in efficient GPU kernels.
>
> Since similar concerns were raised by multiple reviewers, we address the broader question of efficiency analysis in a _general comment_ above. To your specific point, DFloat11 (Zhang et al, 2025) demonstrates that GPU-based Huffman decompression can be implemented with practical performance.
>
> > Variable bit allocation could be explored more and compared to previous works like Evopress.
>
> Thank you for pointing us to this work; we have added the citation and clarified the connection in the revised paper. Since Evopress performs an explicit search over bit allocations, we would expect it to outperform our approach, which incorporates easy-to-calculate sensitivity information to make a one-shot allocation decision.
>
> ---
>
> Thank you for your attention to our responses. If we have managed to alleviate your concerns and address your questions, we kindly ask you to consider updating your score.
>
> .
>
> .
>
> ---
>
> ### References
>
> [Zhang et al, 2025] "70% Size, 100% Accuracy: Lossless LLM Compression for Efficient GPU Inference via Dynamic-Length Float (DFloat11)", Zhang, Tianyi, et al. 2025.

---

> > ### Comment · Reviewer_VQ1o · 2025-11-21
> >
> > "Our main takeaway is that compression schemes that effectively perform variable-length encoding achieve optimal performance under a fixed memory budget."
> >
> > Is this really optimal? What about vector quantization and things like QTIP? I tried calculating entropy for scalar quantization (and doing uneven quantization), but vector quantization was always better.
> >
> > -------
> >
> > Re Dfloat11: That thing is, with all respect to the authors, a bunch of ... They compare performance with respect to dense kernels, but with respect to CPU offloading. Anything is faster than CPU offloading.

---

> > > ### Author Response · Authors · 2025-11-21
> > >
> > > > > "Our main takeaway is that compression schemes that effectively perform variable-length encoding achieve optimal performance under a fixed memory budget."
> > >
> > > > Is this really optimal? What about vector quantization and things like QTIP? I tried calculating entropy for scalar quantization (and doing uneven quantization), but vector quantization was always better.
> > >
> > > A fair question. In terms of a theoretical result, we show this under our assumptions, some of which are quite strong. We understand fixed-length vector quantisation to have two broad capabilities: first is to emulate variable-length scalar codes by trading off size across elements of the vector, second to exploit correlations in the data being represented. The first can be matched by scalar codes (except for some bit-alignment cost), but the second cannot. I believe it is an open question as to how much each of these capabilities contribute to the performance of practical vector codes such as QTIP.
> > >
> > > **VQ results** We have evaluated short, unconstrained VQ, with limited vector length since the number of centroids grows exponentially with this, with the following results:
> > >
> > > | Format | KL (Llama 8B) | KL (3B) | KL (1B) |
> > > | :-: | :-: | :-: | :-: |
> > > | INT + Tensor RMS + Compression | 0.207 | 0.247 | 0.515 |
> > > | VQ[4] + Block Signmax | 0.261 | 0.279 | 0.472 |
> > > | VQ[2] + Block Signmax | 0.299 | 0.423 | 0.582 |
> > > | $\sqrt[3]{p}$ t + Block Signmax | 0.305 | 0.345 | 0.692 |
> > >
> > > Note that the block-signmax variants have 0.25 extra bits per element, due to block scaling overhead. In all cases, VQ can close the gap towards the variable-length scalar code.
> > >
> > > We observed a similar trend for relative RMSE with iid Student’s t data, $\nu=10$ and $b=3$. Scalar Lloyd-Max quantisation achieves $R=0.212$, VQ with $d=2$ has $R=0.186$, $d=4$ has $R=0.161$ and a scalar variable-length code achieves $R=0.150$.
> > >
> > > ---
> > >
> > > > Re Dfloat11: That thing is, with all respect to the authors, a bunch of ... They compare performance with respect to dense kernels, but with respect to CPU offloading. Anything is faster than CPU offloading.
> > >
> > > This is a fair point. In my comment, I was focusing on their numbers from Figure 7, where they can decompress at ~260 GB/s for large inputs, on an A100, achieving an appreciable fraction of the memory bandwidth, 1.5 TB/s (assuming the 40 GB model), although far from saturating it. This is not fast enough to conclude that the formats are already practicable, but is an indication that they may become practicable with future software (and possible hardware) work.

---

> > > > ### Comment · Reviewer_VQ1o · 2025-11-26
> > > >
> > > > Fair enough.
> > > > I think my questions have been answered, I raise my score to 6.

---

### Official Review · Reviewer_9Q1w · 2025-10-28

**Soundness:** 3
**Presentation:** 2
**Contribution:** 2
**Rating:** 4
**Confidence:** 2

**Summary:**

The paper proposes a theoretical framework for choosing the optimal data format for quantization. It first demonstrates that it is sufficient to minimize the squared error between the quantized and non-quantized tensors to find the optimal data format, then compute the optimal format for some known distribution. For unknown distributions, it proposes to fit the experimental distribution with some known ones, using scaling or k-means.
Under this framework, the paper show that variable-length code format consistently outperform fixed-length ones. It is also shown that optimal data format choice can also save up to 0.25 bit per parameter for LLMs.

**Strengths:**

- The paper advances the formalization of quantization data format selection, which is often tackled only empirically;
- Mathematical background seems solid, the SoTA seems adequately cited;
- Supplementary materials is rich and support some asumptions and approximations made in the paper.

**Weaknesses:**

- I found the paper hard to read and follow. Overall structure could be improved. The figures are way out of place with the text. Some figures mentioned in the main text are missing (figure 8, 29, 33...) only to be found in the supplementary material.
- Overall, it seems to me that the main paper is not entirely self-supporting without the help of supplementary material.
- Actual quantization results on LLMs models are completely absent from the paper, and again, can be found only in supplementary material.
- The reduction of the optimal data format problem finding to the minimization of the squared error between quantized and unquantized tensors relies on a lot of approximations. These approximations are further accumuled with the need to fit unknown distributions. Overall, I find it hard to properly appreciate the validity of these, even though some of them are addressed in supplementary material.

**Questions:**

I would appreciate if more insight could be provided regarding the various hypothesis made in the paper. I think that there is both too much information and too little information: the list of all known distribution's optimal format could be shortened. The related work comes very late in the paper with questionable impact. What do the authors think about perhaps simplifying these sections and bringing more insights on the main propositions and demonstrations notably from the supplementary materials?

---

> ### Author Response · Authors · 2025-11-20
>
> Thank you for your constructive review and for highlighting the value of formalising quantisation-format selection, which is the central aim of our work. We greatly appreciate your comments regarding clarity and structure, as they have helped us make improvements to the paper.
>
> ---
>
> > I found the paper hard to read and follow. Overall structure could be improved. The figures are way out of place with the text. Some figures mentioned in the main text are missing (figure 8, 29, 33...) only to be found in the supplementary material.
>
> > Overall, it seems to me that the main paper is not entirely self-supporting without the help of supplementary material.
>
> > I would appreciate if more insight could be provided regarding the various hypothesis made in the paper. I think that there is both too much information and too little information: the list of all known distribution's optimal format could be shortened. The related work comes very late in the paper with questionable impact. What do the authors think about perhaps simplifying these sections and bringing more insights on the main propositions and demonstrations notably from the supplementary materials?
>
> Thank you for these comments. We acknowledge that since the paper attempts to cover the broad design space of formats, the narrative is dense and may be more difficult to follow. Our intention was that the main text supports the core claims regarding variable-length coding and the role of Fisher sensitivity, while the supplementary material provides further evidence (for example in the breadth of models evaluated), and clarifies more subtle technical points (such as the optimal choice of scaling-factor mantissa bits).
>
> Your suggestion to shorten the discussion of constructing optimal formats in order to bring more theoretical and experimental insights in from the supplementary materials is well taken. However, the definitions of block-RMS and block-absmax scaling schemes are critical for understanding Figure 4, showing the effective variable-length coding achieved by absmax scaling. We have thus aimed to improve the flow of Section 2.1 in the updated version of the paper, which will hopefully make the content easier to follow.
>
> > Actual quantization results on LLMs models are completely absent from the paper, and again, can be found only in supplementary material.
>
> While the supplementary material indeed provides further evaluations and results, we include the headline LLM quantisation results in the main body of the paper, mainly, Figures 1, 6 and 7. For downstream task performance, we elected to include averaged task performance shown in Figure 7, as this provided the clearest high-level summary. The overall trend shown in the figure aligns well with the per-task results reported in Table 2 in the supplementary material.
>
> > The reduction of the optimal data format problem finding to the minimization of the squared error between quantized and unquantized tensors relies on a lot of approximations. These approximations are further accumuled with the need to fit unknown distributions. Overall, I find it hard to properly appreciate the validity of these, even though some of them are addressed in supplementary material.
>
> Thank you for highlighting this important point. As multiple reviewers raised similar questions, we have addressed them together in a _general comment_ above. We hope this addresses your concerns.
>
> ---
>
> We hope these clarifications have addressed your main questions. If so, we would greatly appreciate if you could consider increasing your score. Thank you once again for your careful review and the consideration of our rebuttal.

---

### Official Review · Reviewer_HQnU · 2025-11-01

**Soundness:** 3
**Presentation:** 4
**Contribution:** 4
**Rating:** 8
**Confidence:** 4

**Summary:**

This paper addresses the design of optimal quantization formats for neural network weight compression. The authors frame the problem as minimizing KL divergence between original and quantized model outputs under a memory constraint, which they approximate via Fisher-information-weighted squared quantization error. This theoretical reduction enables the application of classical quantization theory techniques to neural network weight compression. They derive optimal element-wise quantizers based on the cube root density rule and extend these to block-scaled formats. A key insight is that effective quantization formats exploit variable-length encoding through either block absmax scaling, sparse outlier storage, or explicit lossless compression. The authors also propose a Fisher-information-based scheme for optimal bit-width allocation across layers. Experiments on multiple LLM families (Llama 3, Qwen 2.5, Gemma 3, Phi 4) validate these insights.

**Strengths:**

- The paper establishes a principled theoretical framework for analyzing quantization formats by reducing the problem to Fisher-information-weighted squared quantization error, enabling the application of classical quantization theory to neural network weight compression. This is a significant, principled contribution that enables systematic format design rather than ad-hoc heuristics.
- The power of this theoretical framework is demonstrated by the authors' ability to directly leverage the rich domain of classical quantization theory to derive concrete technical results: cube root density quantizers for block-scaled Normal, Laplace, and Student-t distributions; the signmax scaling scheme; and a variable bit allocation scheme based on Fisher information. This connection to established theory provides both rigor and a pathway for future advances.
- The variable-length encoding insight provides a unifying explanation for why seemingly disparate techniques (block scaling, sparse outliers, compression) succeed. This is valuable both conceptually---explaining _why_ current methods work---and practically---suggesting that future format designs should explicitly consider variable-length encoding mechanisms.
- The experimental evaluation is comprehensive and rigorous, covering 11 models across 4 families, multiple formats, both direct-cast and QAT settings, and extensive ablations (block size, scale format, symmetric/asymmetric variants). Validation on synthetic data before real models strengthens confidence in the approach.
- The paper is exceptionally well-written with clear progression from problem formulation to theory to empirical validation. Figures effectively communicate key insights and extensive appendices provide implementation details without cluttering the main narrative.
- Quantization is critical for sustainable model training and inference. This paper makes important contributions to our theoretical understanding of quantization---a key step toward developing better methods and providing principled guidance for practitioners.

**Weaknesses:**

- The authors only empirically investigate transformer LLMs, and even among these, Gemma models exhibit behavior that deviates from the theoretical predictions. This raises concerns about the generality of the framework. If discrepancies arise within transformer LLMs alone, it is unclear how well the insights would extend to other architectures such as CNNs, GNNs, or state-space models.
- The theoretical framework relies on three approximations: second-order Taylor expansion of KL divergence, diagonal Fisher approximation, and constant-per-tensor Fisher. The authors provide extensive empirical validations of their assumptions (Figures 10-12) across different models, but it remains unclear whether these approximations hold more generally outside the specific experimental conditions tested.
- As the authors note, even if the cube root density quantizers are theoretically optimal, the practical utility is limited by optimized implementations and hardware support. As the paper's focus is on theoretical insights rather than hardware efficiency, this is not a critical flaw, but it does limit immediate applicability.

**Questions:**

The following questions are intended to help better understand the scope of the work and to think about future directions. These are challenging topics, and a lack of definitive answers is perfectly fine and will not be held against the work.

1. Can you provide more insight into why Gemma models show different behavior compared to Llama, Qwen, and Phi families? What additional analysis or probing have you conducted to understand the source of this discrepancy?
    - Gemma 3 is architecturally distinct from the other models evaluated---it uses a 5:1 local-to-global attention layer split (with different RoPE base frequencies for local and global attention layers) whereas the other models use standard global attention throughout. Do you believe these architectural differences contribute to the observed discrepancies?
    - What do you think the implications of this discrepancy are for the generality of your theoretical framework?
2. The work focuses exclusively on transformer LLMs. How do you anticipate the framework and its underlying assumptions would hold up for fundamentally different architectures, such as CNNs, state-space models, or GNNs? Are there specific architectural features (e.g., different weight distributions, inductive biases) that you believe would make the framework more or less applicable?
3. Can you characterize the regime where your approximations are valid? For instance, under what conditions (model architectures, weight distributions, quantization bit-widths) can practitioners expect any of the three key approximations to break down?

---

> ### Author Response · Authors · 2025-11-20
>
> Thank you for your thorough and positive review of our work on weight compression, and interesting questions regarding the key approximations and behaviour of Gemma models. We have attempted to address these questions below.
>
> ---
>
> > Can you provide more insight into why Gemma models show different behavior compared to Llama, Qwen, and Phi families? What additional analysis or probing have you conducted to understand the source of this discrepancy?
>
> > Gemma 3 is architecturally distinct from the other models evaluated---it uses a 5:1 local-to-global attention layer split (with different RoPE base frequencies for local and global attention layers) whereas the other models use standard global attention throughout. Do you believe these architectural differences contribute to the observed discrepancies?
>
> Thank you for highlighting this. First, a note of correction to the original manuscript: the body text describing Figure 6 correctly stated that Gemma 3 models degraded with variable bit allocation, but the figure did not show this data due to an implementation issue. We have corrected this in the updated version (the new points are marked as “Gemma models”).
>
> To understand the discrepancy, we analysed which parameter tensors correspond to largest differences between actual and predicted KL divergence in Figure 13 (numbering from the updated paper). The largest discrepancies correspond to early layers, where prediction overestimates the effect of perturbations on KL divergence. Further investigation confirmed that the difference is not due to numerical precision issues with `bfloat16` implementation in forward/backward passes. While architectural differences you mention could be an issue, as long as the backward pass is consistent with the forward pass, there is no obvious reason why these would invalidate the framework.
>
> If it were possible, it would be interesting to compare our Fisher estimates with the Adam optimiser’s second-moment state at the end of training (assuming Adam was used), as these should be related. If these agree, an implication of large Fisher information is that early-layer parameters would be frozen towards the end of training.
>
> > What do you think the implications of this discrepancy are for the generality of your theoretical framework?
>
> Without a definitive explanation of the discrepancy, we cannot draw firm conclusions. We are interested in exploring this further, as the question likely relates to optimisation in general, and not just quantisation.
>
> > The authors only empirically investigate transformer LLMs, and even among these, Gemma models exhibit behavior that deviates from the theoretical predictions. This raises concerns about the generality of the framework. If discrepancies arise within transformer LLMs alone, it is unclear how well the insights would extend to other architectures such as CNNs, GNNs, or state-space models.
>
> > The work focuses exclusively on transformer LLMs. How do you anticipate the framework and its underlying assumptions would hold up for fundamentally different architectures, such as CNNs, state-space models, or GNNs? Are there specific architectural features (e.g., different weight distributions, inductive biases) that you believe would make the framework more or less applicable?
>
> None of the assumptions in this work are explicitly architecture-specific. However, almost all parameters in text transformer models are applied as pointwise projections independently across tokens, with the input embedding being the main exception. As you noted, this means that the models we evaluated apply their parameters in similar ways. We did observe that the attention key and value projections, which are shared over query heads in grouped-query attention, are often _more_ sensitive to quantisation (Figure 17). This suggests that models with different resolutions, or stronger parameter sharing, may benefit more from sensitivity-aware quantisation than text transformers. We see this as an exciting direction for future work.
>
> > Can you characterize the regime where your approximations are valid? For instance, under what conditions (model architectures, weight distributions, quantization bit-widths) can practitioners expect any of the three key approximations to break down?
>
> Thank you for raising this question. Since this was raised by multiple reviewers, we have addressed this in a _general comment_ above. While the assumptions behind these approximations do not directly relate to any particular model architecture or weight distribution, the smoothness assumption and basis-dependence of our diagonal-Fisher assumption may indeed vary between model architectures. We agree that further empirical evaluation would provide an interesting extension.
>
> ---
>
> Thank you once again for your comprehensive and encouraging review.

---

> > ### Comment · Reviewer_HQnU · 2025-11-25
> >
> > Thank you for your detailed and comprehensive response. I have no further questions and maintain my original score and positive assessment of the work.

---

### Official Review · Reviewer_ZtPY · 2025-11-05

**Soundness:** 2
**Presentation:** 2
**Contribution:** 2
**Rating:** 2
**Confidence:** 4

**Summary:**

The paper proposes a theoretical framework for designing quantization formats by minimizing the KL divergence between quantized and reference models under a memory constraint. By connecting modern quantization with classical rate–distortion theory, the authors show that efficient schemes perform well because they implicitly use variable length encoding. They derive optimal elementwise quantizers for common distributions, introduce new scaling methods such as RMS, absmax, and signmax, and present a Fisher information based rule for allocating bits across tensors. Experiments on large language models including LLaMA 3, Qwen 2.5, Gemma 3, and Phi 4 show that formats exploiting variable length encoding through block scaling, sparse outlier storage, or compression consistently outperform fixed length ones. The study provides a principled explanation for the effectiveness of formats such as NF4 and SF4 and identifies uniform quantization with lossless compression as the theoretical optimum.

**Strengths:**

The paper provides a solid theoretical perspective by linking neural network quantization with classical information theory, offering useful insights into why certain formats perform well. It introduces new scaling schemes and a Fisher information based bit allocation rule that appear to improve efficiency across model tensors. Experiments on several large language models support the proposed framework and suggest its potential practical value.

**Weaknesses:**

I have the following concerns about the paper:

1. Equations (1) and (2) require stronger justification: Minimizing the KL divergence does not necessarily guarantee that the model’s accuracy will be preserved, and the validity of Equation (2) needs a clearer theoretical explanation.

2. Additional background is needed: to help readers follow the technical development. For instance, the sections around lines 143–146 and 153–157 would benefit from more context and introductory material.

3. Experimental evaluation is insufficient: Given the extensive prior work in this area, it is important to include comparisons with existing quantization methods such as SmoothQuant and QuaRot to better demonstrate the advantages of the proposed approach.

4. This work has limited focus on practical hardware efficiency, as the proposed non-linear quantization formats may be difficult to implement or accelerate on existing hardware.

**Questions:**

How does minimizing KL divergence ensure preservation of task-level accuracy, and could the authors provide theoretical or empirical evidence linking KL divergence to model performance?

What assumptions are required for Equation (2) to hold, and how sensitive are the results to violations of these assumptions?

Can the authors expand the background discussion to better contextualize the derivations around lines 143–157?

---

> ### Author Response · Authors · 2025-11-20
>
> Thank you for your constructive review and for highlighting both the strengths and areas that would benefit from clarification. Your feedback helped us make several improvements in the revised manuscript. We address each point below.
>
> > How does minimizing KL divergence ensure preservation of task-level accuracy, and could the authors provide theoretical or empirical evidence linking KL divergence to model performance?
>
> Thank you for raising this question as it is an important connection to establish. As you note, minimising the KL divergence does not guarantee preservation of task accuracy. However, it is a principled objective with theoretical grounding: note that KL divergence is zero when predictive distributions fully match, implying identical task behaviour. To empirically examine the relationship at non-zero KL values, we added a correlation analysis to the paper (Figure 10). Across model families, we observe a strong correlation between KL divergence (on a generic, pretraining dataset) and downstream task accuracy. The notable exception is Tensor RMS scaling formats, which achieve low KL divergence after QAT without a corresponding improvement in accuracy.
>
> > What assumptions are required for Equation (2) to hold, and how sensitive are the results to violations of these assumptions?
>
> As this helpful question has been raised by multiple reviewers, we have addressed it in the _general comment_ above and have updated Appendix A to clarify the validity of these assumptions.
>
> > Can the authors expand the background discussion to better contextualize the derivations around lines 143–157?
>
> Thank you, we concur that this section required clearer framing and have revised the text and equations to give more context ahead of the definitions.
>
> > Experimental evaluation is insufficient: Given the extensive prior work in this area, it is important to include comparisons with existing quantization methods such as SmoothQuant and QuaRot to better demonstrate the advantages of the proposed approach.
>
> We appreciate this concern. Our primary goal for this work is to evaluate _quantisation formats_, rather than optimisation techniques. For this reason, we focus on direct-cast, which is directly linked to the KL divergence framework and is the simplest quantisation approach, and QAT, which generally provides the strongest results (albeit at higher computational costs). Our QAT results are competitive with those of ParetoQ (Liu et al, 2025), which itself demonstrates the advantage over multiple PTQ techniques (their Table 4, c.f. our Table 2). Due to the overall consistency of our direct-cast and QAT results, we argue that our existing experimental analysis gives a strong indication of the comparative performance of different quantisation formats. We agree that validating the results under PTQ optimisation would be interesting further work, and have added a note indicating this.
>
> > This work has limited focus on practical hardware efficiency, as the proposed non-linear quantization formats may be difficult to implement or accelerate on existing hardware.
>
> This important point was raised by several reviewers, and we have addressed it in the _general comment_ above. We hope this addresses your concerns about the applicability of this work.
>
> ---
>
> We hope these clarifications and revisions strengthen your confidence in the soundness and relevance of our work. If so, we would greatly appreciate if you considered increasing your score. Thank you once again for your thoughtful review and taking the time to evaluate our rebuttal.
>
> .
>
> .
>
> ---
>
> ### References
>
> [Liu et al, 2025] "ParetoQ: Scaling laws in extremely low-bit LLM quantization." Liu, Zechun, et al., 2025.

---

### Author Response · Authors · 2025-11-20
**General comment**

We thank all reviewers for their thoughtful feedback. We are encouraged that several reviewers (ZtPY, HQnU, 9Q1w) found our formalisation and foundations solid and principled, and that our empirical analysis was viewed as thorough and rigorous (HQnU, VQ1o). We identified two primary concerns that we address below, with further points covered in the individual responses.

We have updated the manuscript in response to the reviewers’ questions and suggestions, which we greatly appreciate.

.

---
### Approximations (ZtPY, HQnU, 9Q1w)

_Can we confirm the validity of the approximations required to link KL divergence to minimising squared error?_

The three key approximations are: 1) a second-order Taylor expansion of KL divergence, 2) diagonal Fisher and 3) constant-per-tensor Fisher.

Before discussing them individually, we note that these approximations are **only used to inform format design** (in our notation, choosing the set $\tilde{\Theta}$), but not to search for optimal quantised parameters within these formats. Applying them to optimisation would imply that direct-cast (round-to-nearest) is optimal, however this is known to perform poorly compared with post-training quantisation and quantisation-aware training approaches. This is a subtle, but an important distinction.

We also compare these approximations to the local-objective assumption, commonly underlying PTQ techniques such as GPTQ (Frantar et al., 2023). These methods replace the global optimisation objective (such as output KL divergence) with a local layer-wise squared reconstruction error, allowing the use of full (per-layer) Hessian, but relying on the strong assumption that improving the local objective is aligned with improving the global objective. As with SqueezeLLM (Kim et al, 2024), our method maintains a global objective; thus, relative strength of assumptions is not strictly comparable.

We have added additional discussion sections for each approximation into Appendix A to further guide the reader, which we summarise here:

 - **2nd order expansion**: Assumes smoothness and a small perturbations. Smoothness is typically expected, as the models were trained with this assumption. Perturbation magnitude is hard to evaluate, but this effect will make the approximation worse for low bit-width formats.

 - **Diagonal Fisher**: A strong assumption which is not robust to a change of basis (e.g. if optimising rotated parameters), but still commonly used due to tractability (e.g. Adam optimiser).

 - **Scaled-identity Fisher**: Empirically, diagonal Fisher values vary both within and across tensors (Figure 12). We use this assumption only in specific cases (allocating variable bit-width per tensor), but not throughout the work; formats such as weighted Lloyd-Max do not rely on it.

---

### Hardware efficiency: (ZtPY, HQnU, VQ1o)

_Not all formats considered in the work permit efficient implementation._

We agree that not all formats considered permit efficient hardware implementation (as acknowledged in Section 6). Our aim, however, was to characterise the **space of possible size vs. KL divergence tradeoffs**, which includes formats that may be impractical today. The scope is intentional for several reasons:

 - Entropy-coded formats establish an upper bound on achievable compression performance; they showcase a gap between current hardware-efficient formats and the theoretical frontier.

 - There already exist demonstrations of fast nonlinear formats and entropy codes. For example, QLoRA (Dettmers et al, 2023) reports speed claims while using the nonlinear NF4 format, and DFloat11 (Zhang et al, 2025) employs efficient Huffman coding to compress the exponents of bfloat16 tensors.

 - A comprehensive study of runtime performance across all linear/nonlinear and grouped, sparse, or compression schemes would undoubtedly be valuable, but is beyond the scope of current work, which focuses on the compression limits of quantisation formats.

.

.

---

### References

[Frantar et al., 2023] "GPTQ: Accurate post-training quantization for generative pre-trained transformers", Frantar, Elias, et al., 2022.

[Kim et al, 2024] "SqueezeLLM: Dense-and-sparse quantization”, Kim, Sehoon, et al. 2024.

[Dettmers et al, 2023] "QLoRA: Efficient finetuning of quantized LLMs", Dettmers, Tim, et al., 2023.

[Zhang et al, 2025] "70% Size, 100% Accuracy: Lossless LLM Compression for Efficient GPU Inference via Dynamic-Length Float (DFloat11)", Zhang, Tianyi, et al. 2025.

---

### Meta-Review · Area_Chair_wrPj · 2026-01-03

**Summary:**

This paper proposes a principled framework for **designing and analyzing weight-quantization formats** by framing format selection as minimizing **output KL divergence** under a **model-size (bits/parameter) constraint**, then approximating this objective with a **Fisher-weighted squared quantization error** to connect modern LLM quantization with classical rate–distortion / entropy-constrained quantization theory.

Through experiments across multiple transformer LLM families (e.g., Llama/Qwen/Gemma/Phi), the authors argue that the strongest practical formats effectively behave like **variable-length encoders**, via block scaling, sparse outliers, or explicit compression—and use the framework to derive nonlinear curves and a Fisher-based **bit-allocation rule** that can reduce average bitwidth.

I recommend **Reject**: despite a compelling high-level narrative and broad LLM coverage, the decision-critical concerns likely remain unresolved for enough reviewers, namely (i) the reliance on **stacked, strong approximations** to justify “optimality,” (ii) a **weak/indirect link** from KL minimization to downstream task quality across settings, (iii) **scope-limited comparisons** to optimization-centric PTQ baselines many reviewers expect, and (iv) limited immediate **hardware practicality** for several nonlinear/compression-heavy formats (even if intentionally out of scope).

**Reviewer Concerns:**

* **Key approximations are too strong / unclear validity of the KL → squared-error (Fisher-weighted) reduction** *(ZtPY, 9Q1w, HQnU)*: **Partially addressed (likely).** The rebuttal expands discussion of the three approximations and adds empirical checks/correlation analyses, which likely satisfies **HQnU** and helps **9Q1w**, but **ZtPY** would likely still view Eq.(1)/(2) justification as not fully convincing (especially at low bit-widths where perturbations are not “small”).

* **Objective mismatch: minimizing KL does not guarantee task accuracy preservation** *(ZtPY)*: **Partially addressed (likely).** Added evidence that KL correlates with downstream accuracy helps, but the rebuttal also concedes it is not a guarantee and shows exceptions (e.g., certain scaling variants). I expect **ZtPY** would still treat this as a central limitation.

* **Missing/insufficient comparisons to common PTQ optimization baselines (e.g., SmoothQuant, QuaRot) and “apples-to-apples” evaluation** *(ZtPY)*: **Likely still outstanding.** The rebuttal argues the paper is about *formats* rather than *optimization methods* and points indirectly to other work (e.g., ParetoQ) for the PTQ context. I expect this would not fully satisfy a reviewer asking for direct head-to-head comparisons.

* **Practical hardware efficiency is unclear for nonlinear / entropy-coded / compression-heavy formats** *(ZtPY, VQ1o)*: **Partially addressed (likely).** The rebuttal clarifies this is partly out of scope and cites examples suggesting feasibility, but skepticism likely remains about efficient GPU kernels and end-to-end throughput for many studied formats.

* **Presentation/readability and “main takeaway” clarity (paper dense; results/figures hard to follow; main message feels diffuse)** *(9Q1w, VQ1o)*: **Partially addressed (likely).** The rebuttal claims structural edits and clearer framing, which may alleviate **9Q1w** somewhat, but **VQ1o**’s concern about the overarching message (and practicality) likely remains a risk for keeping the paper below threshold.

**Reviewer Scores:**

## Reviewer HQnU: 8 → 8
Why: This reviewer was already strongly positive (theory/principled framing + rigorous empirical coverage) and explicitly indicated post-rebuttal that they had no further questions and would **maintain the original score**.

## Reviewer VQ1o: 4 → 6
Why: Their initial concerns were “unclear main takeaway,” skepticism about the variable-length-encoding claim, and practicality of Huffman-style ideas. The rebuttal/discussion directly engaged these points (including the scalar-vs-vector clarification and additional quantitative comparisons/intuition), and the reviewer explicitly stated post-rebuttal that their questions were answered and they **raise the score to 6**.

## Reviewer 9Q1w: 4 → 4
Why: The review is dominated by **presentation/structure** issues (main paper not self-contained; figures/results perceived as missing or pushed to the supplement) and skepticism about the *stack of approximations*. The rebuttal acknowledges these and claims improved flow plus more discussion in the appendix, but the core concern is that the *main paper narrative* remains hard to follow. I therefore expect this reviewer would likely **stay at 4** (possibly more positive in tone, but not enough to jump a discrete level).

## Reviewer ZtPY: 2 → 4
Why: The negative score is driven by (i) insufficient justification for the KL→squared-error link (Eq. 1–2), (ii) missing context/background around key derivations, (iii) missing comparisons to PTQ methods like SmoothQuant/QuaRot, and (iv) hardware implementability concerns for nonlinear formats. The rebuttal helps on (i) and (ii) by explicitly enumerating assumptions, adding appendix discussion, and adding an empirical KL↔accuracy correlation analysis; it also partially addresses (iv) by clarifying scope and citing feasibility examples. However, (iii) remains mostly a scope deflection (format study vs PTQ optimization), and hardware practicality is still not resolved. Net: I’d expect a move from **2 → 4** (still below threshold).

---

### Decision · Program_Chairs · 2026-01-26

Reject